# Improved Particle Approximation Error for Mean Field Neural Networks

**Atsushi Nitanda**

CFAR and IHPC, Agency for Science, Technology and Research (A⋆STAR), Singapore
College of Computing and Data Science, Nanyang Technological University, Singapore
`atsushi_nitanda@cfar.a-star.edu.sg`

## Abstract

Mean-field Langevin dynamics (MFLD) minimizes an entropy-regularized nonlinear convex functional defined over the space of probability distributions. MFLD has gained attention due to its connection with noisy gradient descent for mean-field two-layer neural networks. Unlike standard Langevin dynamics, the nonlinearity of the objective functional induces particle interactions, necessitating multiple particles to approximate the dynamics in a finite-particle setting. Recent works (Chen et al., 2022; Suzuki et al., 2023b) have demonstrated the *uniform-in-time propagation of chaos* for MFLD, showing that the gap between the particle system and its mean-field limit uniformly shrinks over time as the number of particles increases. In this work, we improve the dependence on logarithmic Sobolev inequality (LSI) constants in their particle approximation errors which can exponentially deteriorate with the regularization coefficient. Specifically, we establish an LSI-constant-free particle approximation error concerning the objective gap by leveraging the problem structure in risk minimization. As the application, we demonstrate improved convergence of MFLD, sampling guarantee for the mean-field stationary distribution, and uniform-in-time Wasserstein propagation of chaos in terms of particle complexity.

## 1 Introduction

In this work, we consider the following entropy-regularized mean-field optimization problem:

$$\mathcal{L}(\mu) = F(\mu) + \lambda \mathrm{Ent}(\mu), \tag{1}$$

where $F : \mathcal{P}_2(\mathbb{R}^d) \to \mathbb{R}$ is a convex functional on the space of probability distributions $\mathcal{P}_2(\mathbb{R}^d)$ and $\mathrm{Ent}(\mu) = \int \mu(\mathrm{d}x) \log \frac{\mathrm{d}\mu}{\mathrm{d}x}(x)$ is a negative entropy. Especially we focus on the learning problem of mean-field neural networks, that is, $F(\mu)$ is a risk of (infinitely wide) two-layer neural networks $\mathbb{E}_{X \sim \mu}[h(X, \cdot)]$ where $h(X, \cdot)$ represents a single neuron with parameter $X$. One advantage of this problem is that the convexity of $F$ with respect to $\mu$ can be leveraged to analyze gradient-based methods for a finite-size two-layer neural network: $\frac{1}{N} \sum_{i=1}^{N} h(x^i, \cdot) \ (x^i \in \mathbb{R}^d)$. This is achieved by translating the optimization dynamics of the finite-dimensional parameters $(x^1, \ldots, x^N) \in \mathbb{R}^{dN}$ into the dynamics of $\mu$ via the mean-field limit: $\frac{1}{N} \sum_{i=1}^{N} \delta_{x^i} \to \mu \ (N \to \infty)$. This connection was pointed out by Nitanda and Suzuki (2017); Mei et al. (2018); Chizat and Bach (2018); Rotskoff and Vanden-Eijnden (2022); Sirignano and Spiliopoulos (2020b,a) in the case of $\lambda = 0$, and used for showing the global convergence of the gradient flow for (1) by Mei et al. (2018); Chizat and Bach (2018).

One may consider adding Gaussian noise to the gradient descent to make the method more stable. Then, we arrive at the following *mean-field Langevin dynamics* (MFLD) (Hu et al., 2019; Mei et al., 2018) as a continuous-time representation under $N = \infty$ of this noisy gradient descent.

$$\mathrm{d}X_t = -\nabla \frac{\delta F(\mu_t)}{\delta \mu}(X_t)\mathrm{d}t + \sqrt{2\lambda}\mathrm{d}W_t, \quad \mu_t = \mathrm{Law}(X_t), \tag{2}$$

where $\{W_t\}_{t\geq 0}$ is the $d$-dimensional standard Brownian motion and $\nabla\frac{\delta F(\mu)}{\delta\mu}$ is the Wasserstein gradient that is the gradient of the first-variation $\frac{\delta F(\mu)}{\delta\mu}$ of $F$. Even though several optimization methods (Nitanda et al., 2021; Oko et al., 2022; Chen et al., 2023) that can efficiently solve the above problem with polynomial computational complexity have been proposed, MFLD remains an interesting research subject because of the above connection to the noisy gradient descent. In fact, recent studies showed that MFLD globally converges to the optimal solution (Hu et al., 2019; Mei et al., 2018) thanks to noise perturbation and that its convergence rate is exponential in continuous-time under *uniform log-Sobolev inequality* (Nitanda et al., 2022; Chizat, 2022).

However, despite such remarkable progress, the above studies basically assume the mean-field limit: $N = \infty$. To analyze an implementable MFLD, we have to deal with discrete-time and finite-particle dynamics, i.e., noisy gradient descent:

$$X_{k+1}^i = X_k^i - \eta\nabla\frac{\delta F(\mu_{\mathbf{X}_k})}{\delta\mu}(X_k^i) + \sqrt{2\lambda\eta}\xi_k^i, \quad (i \in \{1,\dots,N\}), \tag{3}$$

where $\xi_k^i \sim \mathcal{N}(0, I_d)$ $(i \in \{1,\dots,d\})$ are i.i.d. standard normal random variables and $\mu_{\mathbf{X}_k} = \frac{1}{N}\sum_{i=1}^N \delta_{X_k^i}$ is an empirical measure. On the one hand, the convergence in the discrete-time setting has been proved by Nitanda et al. (2022) using the one-step interpolation argument for Langevin dynamcis (Vempala and Wibisono, 2019). On the other hand, approximation error induced by using finite-particle system $\mathbf{X}_k = (X_k^1, \dots, X_k^N)$ has been studied in the literature of *propagation of chaos* (Sznitman, 1991). As for MFLD, Mei et al. (2018) suggested exponential blow-up of particle approximation error in time, but recent works (Chen et al., 2022; Suzuki et al., 2023a) proved *uniform-in-time propagation of chaos*, saying that the gap between $N$-particle system and its mean-field limit shrinks uniformly in time as $N \to \infty$. Afterward, Suzuki et al. (2023b) established truly quantitative convergence guarantees for (3) by integrating the techniques developed in Nitanda et al. (2022); Chen et al. (2022). Furthermore, Kook et al. (2024) proved the sampling guarantee for the mean-field stationary distribution: $\mu_* = \arg\min_{\mathcal{P}_2(\mathbb{R}^d)} \mathcal{L}(\mu)$, building upon the uniform-in-time propagation of chaos.

## 1.1 Contributions

In this work, we further improve the particle approximation error (Chen et al., 2022; Suzuki et al., 2023b) by alleviating the dependence on logarithmic Sobolev inequality (LSI) constants in their bounds. This improvement could exponentially reduce the required number of particles because LSI constant $\alpha$ could exponentially deteriorate with the regularization coefficient, i.e., $\alpha \gtrsim \exp(-\Theta(1/\lambda))$. Specifically, we establish an LSI-constant-free particle approximation error concerning the objective gap by leveraging the problem structure in risk minimization. Additionally, as the application, we demonstrate improved (i) convergence of MFLD, (ii) sampling guarantee for the mean-field stationary distribution $\mu_*$, and (iii) uniform-in-time Wasserstein propagation of chaos in terms of particle complexity. We summarize our contributions below.

- We demonstrate the particle approximation error $O(\frac{1}{N})$ (Theorem 1) regarding the objective gap. A significant difference from the existing approximation error $O(\frac{\lambda}{\alpha N})$ (Chen et al., 2022; Suzuki et al., 2023b) is that our bound is free from the LSI-constant. Therefore, the approximation error uniformly decreases as $N \to \infty$ regardless of the value of LSI-constant as well as $\lambda$.

- As applications of Theorem 1, we derive the convergence rates of the finite-particle MFLDs (Theorem 2), sampling guarantee for $\mu_*$ (Corollary 1), and uniform-in-time Wasserstein propagation of chaos (Corollary 2) with the approximation errors inherited from Theorem 1, which improve upon existing errors (Chen et al., 2022; Suzuki et al., 2023b; Kook et al., 2024).

Here, we mention the proof strategy of Theorem 1. Langevin dynamics (LD) is a special case of MFLD corresponding to the case where $F$ is a linear functional. It is well known that even with a single particle, we can simulate LD and the particle converges to the target Gibbs distribution. This means that the particle approximation error is due to the non-linearity of $F$. Therefore, in our analysis, we carefully treat the non-linearity of $F$ and obtain an expression for the particle approximation error using the Bregman divergence induced by $F$. Finally, we relate this divergence to the variance of an $N$-particle neural network and show the error of $O(1/N)$. This proof strategy is quite different from existing ones and is simple. Moreover, it leads to an improved approximation error as mentioned above. We refer the readers to Section 4 for details about the proof.

## 1.2 Notations

We denote vectors and random variables on $\mathbb{R}^d$ by lowercase and uppercase letters such as $x$ and $X$, respectively, and boldface is used for $N$-pairs of them like $\mathbf{x} = (x^1, \ldots, x^N) \in \mathbb{R}^{Nd}$ and $\mathbf{X} = (X^1, \ldots, X^N)$. $\|\cdot\|_2$ denotes the Euclidean norm. Let $\mathcal{P}_2(\mathbb{R}^d)$ be the set of probability distributions with finite second moment on $\mathbb{R}^d$. For probability distributions $\mu, \nu \in \mathcal{P}_2(\mathbb{R}^d)$, we define Kullback-Leibler (KL) divergence (a.k.a. relative entropy) by $\mathrm{KL}(\mu\|\nu) \overset{\text{def}}{=} \int \mathrm{d}\mu(x) \log \frac{\mathrm{d}\mu}{\mathrm{d}\nu}(x)$. Ent denotes the negative entropy: $\mathrm{Ent}(\mu) = \int \mu(\mathrm{d}x) \log \frac{\mathrm{d}\mu}{\mathrm{d}x}(x)$. We denote by $\frac{\mathrm{d}\mu}{\mathrm{d}x}$ the density function of $\mu$ with respect to the Lebesgue measure if it exists. We denote $\langle f, m \rangle = \int f(x)m(\mathrm{d}x)$ for a (singed) measure $m$ and integrable function $f$ on $\mathbb{R}^d$. Given $\mathbf{x} = (x^1, \ldots, x^N) \in \mathbb{R}^{Nd}$, we write an empirical measure supported on $\mathbf{x}$ as $\mu_{\mathbf{x}} = \frac{1}{N} \sum_{i=1}^{N} \delta_{x^i}$.

# 2 Preliminaries

In this section, we explain a problem setting and give a brief overview of the recent progress of the mean-field Langevin dynamics.

## 2.1 Problem setting

We say the functional $G : \mathcal{P}_2(\mathbb{R}^d) \to \mathbb{R}$ is differentiable when there exists a functional (referred to as a *first variation*): $\frac{\delta G}{\delta \mu} : \mathcal{P}_2(\mathbb{R}^d) \times \mathbb{R}^d \ni (\mu, x) \mapsto \frac{\delta G(\mu)}{\delta \mu}(x) \in \mathbb{R}$ such that for $\forall \mu, \mu' \in \mathcal{P}_2(\mathbb{R}^d)$,

$$\left.\frac{\mathrm{d}G(\mu + \epsilon(\mu' - \mu))}{\mathrm{d}\epsilon}\right|_{\epsilon=0} = \int \frac{\delta G(\mu)}{\delta \mu}(x)(\mu' - \mu)(\mathrm{d}x),$$

and say $G$ is convex when for $\forall \mu, \mu' \in \mathcal{P}_2(\mathbb{R}^d)$,

$$G(\mu') \geq G(\mu) + \int \frac{\delta G(\mu)}{\delta \mu}(x)(\mu' - \mu)(\mathrm{d}x). \tag{4}$$

For a differentiable and convex functional $F_0 : \mathcal{P}_2(\mathbb{R}^d) \to \mathbb{R}$ and coefficients $\lambda$, $\lambda' > 0$ we consider the minimization of an entropy-regularized convex functional (Mei et al., 2018; Hu et al., 2019; Nitanda et al., 2022; Chizat, 2022; Chen et al., 2022; Suzuki et al., 2023b; Kook et al., 2024):

$$\min_{\mu \in \mathcal{P}_2(\mathbb{R}^d)} \left\{ \mathcal{L}(\mu) = F_0(\mu) + \lambda' \mathbb{E}_{X \sim \mu}[\|X\|_2^2] + \lambda \mathrm{Ent}(\mu) \right\}. \tag{5}$$

We set $F(\mu) = F_0(\mu) + \lambda' \mathbb{E}_\mu[\|X\|_2^2]$. Note both $F$ and $\mathcal{L}$ are convex functionals. In particular, we focus on the empirical risk $F_0$ of the mean-field neural networks, i.e., two-layer neural networks in the mean-field regime. The definition of this model is given in Section 3. Throughout the paper, we assume the existence of the solution $\mu_* \in \mathcal{P}_2(\mathbb{R}^d)$ of the problem (5) and make the following regularity assumption on the objective function, which is inherited from Chizat (2022); Nitanda et al. (2022); Chen et al. (2023).

**Assumption 1.** *There exists $M_1, M_2 > 0$ such that for any $\mu \in \mathcal{P}_2(\mathbb{R}^d)$, $x \in \mathbb{R}^d$, $\left|\nabla \frac{\delta F_0(\mu)}{\delta \mu}(x)\right| \leq M_1$ and for any $\mu, \mu' \in \mathcal{P}_2(\mathbb{R}^d)$, $x, x' \in \mathbb{R}^d$,*

$$\left\|\nabla \frac{\delta F_0(\mu)}{\delta \mu}(x) - \nabla \frac{\delta F_0(\mu')}{\delta \mu}(x')\right\|_2 \leq M_2 \left(W_2(\mu, \mu') + \|x - x'\|_2\right).$$

Then, under Assumption 1, $\mu_*$ uniquely exists and satisfies the optimality condition: $\mu_* \propto \exp\left(-\frac{1}{\lambda} \frac{\delta F(\mu_*)}{\delta \mu}\right)$. We refer the readers to Chizat (2022); Hu et al. (2019); Mei et al. (2018) for details.

We introduce the *proximal Gibbs distribution* (Nitanda et al., 2022; Chizat, 2022), which plays a key role in showing the convergence of mean-field optimization methods (Nitanda et al., 2022; Chizat, 2022; Oko et al., 2022; Chen et al., 2023).

**Definition 1** (Proximal Gibbs distribution). *For $\mu \in \mathcal{P}_2(\mathbb{R}^d)$, the proximal Gibbs distribution $\hat{\mu}$ associated with $\mu$ is defined as follows:*

$$\frac{\mathrm{d}\hat{\mu}}{\mathrm{d}x}(x) = \frac{\exp\left(-\frac{1}{\lambda}\frac{\delta F(\mu)}{\delta\mu}(x)\right)}{Z(\mu)}, \tag{6}$$

*where $Z(\mu)$ is the normalization constant and $\mathrm{d}\hat{\mu}/\mathrm{d}x$ is the density function w.r.t. Lebesgue measure.*

We remark that $\hat{\mu}$ exists, that is $Z(\mu) < \infty$, for any $\mu \in \mathcal{P}_2(\mathbb{R}^d)$ because of the boundedness of $\delta F_0/\delta\mu$ in Assumption 1 and that the optimality condition for the problem (5) can be simply written using $\hat{\mu}$ as follows: $\mu_* = \hat{\mu}_*$. Since the proximal Gibbs distribution $\hat{\mu}$ minimizes the linear approximation of $F$ at $\mu$: $F(\mu) + \int \frac{\delta F}{\delta\mu}(\mu)(x)(\mu' - \mu)(\mathrm{d}x) + \lambda\mathrm{Ent}(\mu')$ with respect to $\mu'$, $\hat{\mu}$ can be regarded as a surrogate of the solution $\mu_*$. In the case where $F_0(\mu)$ is a linear functional: $F_0(\mu) = \mathbb{E}_\mu[f]$ ($\exists f : \mathbb{R}^d \to \mathbb{R}$), the proximal Gibbs distribution $\hat{\mu}$ coincides with $\mu_*$.

## 2.2 Mean-field Langevin dynamics and finite-particle approximation

The mean field Langevin dynamics (MFLD) is one effective method for solving the problem (5). MFLD $\{X_t\}_{t\geq 0}$ is described by the following stochastic differential equation:

$$\mathrm{d}X_t = -\nabla\frac{\delta F}{\delta\mu}(\mu_t)(X_t)\mathrm{d}t + \sqrt{2\lambda}\mathrm{d}W_t, \quad \mu_t = \mathrm{Law}(X_t), \tag{7}$$

where $\{W_t\}_{t\geq 0}$ is the $d$-dimensional standard Brownian motion with $W_0 = 0$. We refer the reader to Huang et al. (2021) for the existence of the unique solution of this equation under Assumption 1. Nitanda et al. (2022); Chizat (2022) showed the convergence of MFLD: $\mathcal{L}(\mu_t) - \mathcal{L}(\mu_*) \leq \exp(-2\alpha\lambda t)(\mathcal{L}(\mu_0) - \mathcal{L}(\mu_*))$ under the *uniform log-Sobolev inequality (LSI)*:

**Assumption 2.** *There exists a constant $\alpha > 0$ such that for any $\mu \in \mathcal{P}_2(\mathbb{R}^d)$, proximal Gibbs distribution $\hat{\mu}$ satisfies log-Sobolev inequality with $\alpha$, that is, for any smooth function $g : \mathbb{R}^d \to \mathbb{R}$,*

$$\mathbb{E}_{\hat{\mu}}[g^2 \log g^2] - \mathbb{E}_{\hat{\mu}}[g^2] \log \mathbb{E}_{\hat{\mu}}[g^2] \leq \frac{2}{\alpha}\mathbb{E}_{\hat{\mu}}[\|\nabla g\|_2^2].$$

Because of the appearance of $\mu_t$ in the drift term, MFLD is a distribution-dependent dynamics referred to as general McKean–Vlasov (McKean Jr, 1966). This dependence makes the difference from the standard Langevin dynamics. Hence, we need multiple particles to approximately simulate MFLD (7) unlike the standard Langevin dynamics. We here introduce the finite-particle approximation of (7) described by the $N$-tuple of stochastic differential equation $\{\mathbf{X}_t\}_{t\geq 0} = \{(X_t^1, \ldots, X_t^N)\}_{t\geq 0}$:

$$\mathrm{d}X_t^i = -\nabla\frac{\delta F(\mu_{\mathbf{X}_t})}{\delta\mu}(X_t^i)\mathrm{d}t + \sqrt{2\lambda}\mathrm{d}W_t^i, \quad (i \in \{1, \ldots, N\}), \tag{8}$$

where $\mu_{\mathbf{X}_t} = \frac{1}{N}\sum_{i=1}^N \delta_{X_t^i}$ is an empirical measure supported on $\mathbf{X}_t$, $\{W_t^i\}_{t\geq 0}$, ($i \in \{1, \ldots, N\}$) are independent standard Brownian motions, and the gradient in the first term in RHS is taken for the function: $\frac{\delta F(\mu_{\mathbf{X}_t})}{\delta\mu}(\cdot) : \mathbb{R}^d \to \mathbb{R}$. We often denote $F(\mathbf{x}) = F(\mu_{\mathbf{x}})$ when emphasizing $F$ as a function of $\mathbf{x}$. Noticing $N\nabla_{x^i}F(\mathbf{x}) = \nabla\frac{\delta F(\mu_{\mathbf{x}})}{\delta\mu}(x^i)$ (Chizat, 2022), we can identify the dynamics (8) as the Langevin dynamics $\mathrm{d}\mathbf{X}_t = -N\nabla F(\mathbf{X}_t)\mathrm{d}t + \sqrt{2\lambda}\mathrm{d}\mathbf{W}_t$, where $\{\mathbf{W}_t\}_{t\geq 0}$ is the standard Brownian motion on $\mathbb{R}^{dN}$, for sampling from the following Gibbs distribution $\mu_*^{(N)}$ on $\mathbb{R}^{dN}$ (Chen et al., 2022):

$$\frac{\mathrm{d}\mu_*^{(N)}}{\mathrm{d}\mathbf{x}}(\mathbf{x}) \propto \exp\left(-\frac{N}{\lambda}F(\mathbf{x})\right) = \exp\left(-\frac{N}{\lambda}F_0(\mathbf{x}) - \frac{\lambda'}{\lambda}\|\mathbf{x}\|_2^2\right). \tag{9}$$

In other words, the dynamics (8) minimizes the entropy-regularized linear functional: $\mu^{(N)} \in \mathcal{P}_2(\mathbb{R}^{dN})$,

$$\mathcal{L}^{(N)}(\mu^{(N)}) = N\mathbb{E}_{\mathbf{X}\sim\mu^{(N)}}[F(\mathbf{X})] + \lambda\mathrm{Ent}(\mu^{(N)}), \tag{10}$$

and $\mu_*^{(N)}$ is the minimizer of $\mathcal{L}^{(N)}$. Therefore, two objective functions $\mathcal{L}$ and $\mathcal{L}^{(N)}$ are tied together through the two aspects of the dynamics (8); one is the finite-particle approximation of the MFLD (7)

for $\mathcal{L}$ and the other is the optimization methods for $\mathcal{L}^{(N)}$. We then expect $\mathcal{L}^{(N)}(\mu_*^{(N)})/N$ converges to $\mathcal{L}(\mu_*)$ as $N \to \infty$. Such finite-particle approximation error between $\mathcal{L}^{(N)}(\mu_*^{(N)})/N$ and $\mathcal{L}(\mu_*)$ has been studied in the literature of *propagation of chaos*. Especially, Chen et al. (2022) proved

$$\frac{\lambda}{N}\mathrm{KL}(\mu_*^{(N)}\|\mu_*^{\otimes N}) \le \frac{1}{N}\mathcal{L}^{(N)}(\mu_*^{(N)}) - \mathcal{L}(\mu_*) \le \frac{\lambda C}{\alpha N} \tag{11}$$

where $C > 0$ is some constant and $\mu_*^{\otimes N}$ is an $N$-product measure of $\mu_*$. Suzuki et al. (2023b) further studied MFLD in finite-particle and discrete-time setting defined below: given $k$-th iteration $\mathbf{X}_k = (X_k^1, \ldots, X_k^N)$,

$$X_{k+1}^i = X_k^i - \eta\nabla\frac{\delta F(\mu_{\mathbf{X}_k})}{\delta\mu}(X_k^i) + \sqrt{2\lambda\eta}\xi_k^i, \quad (i \in \{1, \ldots, N\}), \tag{12}$$

where $\xi_k^i \sim \mathcal{N}(0, I_d)$ $(i \in \{1, \ldots, N\})$ are i.i.d. standard normal random variables. By extending the proof techniques developed by Chen et al. (2022), Suzuki et al. (2023b) proved the uniform-in-time propagation of chaos for MFLD (12); there exist constants $C_1, C_2 > 0$ such that

$$\frac{1}{N}\mathcal{L}^{(N)}(\mu_k^{(N)}) - \mathcal{L}(\mu_*) \le \exp\left(-\lambda\alpha\eta k/2\right)\left(\frac{1}{N}\mathcal{L}^{(N)}(\mu_0^{(N)}) - \mathcal{L}(\mu_*)\right) + \frac{(\lambda\eta + \eta^2)C_1}{\lambda\alpha} + \frac{\lambda C_2}{\alpha N}, \tag{13}$$

where $\mu_k^{(N)} = \mathrm{Law}(\mathbf{X}_k)$. The last two terms are due to time-discretization and finite-particle approximation, respectively. The finite-particle approximation error $O(\frac{\lambda}{\alpha N})$ appearing in (11), (13) means the deterioration as $\alpha \to 0$. Considering typical estimation $\alpha \gtrsim \exp(-\Theta(1/\lambda))$ (e.g., Theorem 1 in Suzuki et al. (2023b)) of LSI-constant using Holley and Stroock argument (Holley and Stroock, 1987) or Miclo's trick (Bardet et al., 2018), these bounds imply that the required number of particles increases exponentially as $\lambda \to 0$.

## 3 Main results

In this section, we present an LSI-constant free particle approximation error between $\frac{1}{N}\mathcal{L}^{(N)}(\mu_*^{(N)})$ and $\mathcal{L}(\mu_*)$ for mean-field neural networks and apply it to the mean-field Langevin dynamics.

### 3.1 LSI-constant free particle approximation error for mean-field neural networks

We focus on the empirical risk minimization problem of mean-field neural networks. Let $h(x, \cdot) : \mathcal{Z} \to \mathbb{R}$ be a function parameterized by $x \in \mathbb{R}^d$, where $\mathcal{Z}$ is the data space. The mean-field model is obtained by integrating $h(x, \cdot)$ with respect to the probability distribution $\mu \in \mathcal{P}_2(\mathbb{R}^d)$ over the parameter space: $h_\mu(\cdot) = \mathbb{E}_{X\sim\mu}[h(X, \cdot)]$. Typically, $h$ is set as $h(x, z) = \sigma(w^\top z)$ or $h(x, z) = \tanh(v\sigma(w^\top z))$ where $\sigma$ is an activation function and $x = w$ or $x = (v, w)$ is the trainable parameter in each case. Given training examples $\{(z_j, y_j)\}_{j=1}^n \subset \mathcal{Z} \times \mathbb{R}$ and loss function $\ell(\cdot, \cdot) : \mathbb{R} \times \mathbb{R} \to \mathbb{R}$, we consider the empirical risk of the mean-field neural networks:

$$F_0(\mu) = \frac{1}{n}\sum_{j=1}^n \ell(h_\mu(z_j), y_j). \tag{14}$$

For our analysis, we make the following assumption which is satisfied in the common settings.

**Assumption 3.** $\ell(\cdot, y)$ *is convex and $L$-smooth, and $h(X, z)$ $(X \sim \mu_*)$ has a finite-second moment;*

- *There exists $L > 0$ such that for any $a, b, y \in \mathcal{Y}$, $\ell(b, y) \le \ell(a, y) + \frac{\partial\ell(a,y)}{\partial a}(b - a) + \frac{L}{2}|b - a|^2$.*

- *There exists $R > 0$ such that for any $z \in \mathcal{Z}$, $\mathbb{E}_{X\sim\mu_*}[|h(X, z)|^2] \le R^2$.*

We can directly verify this assumption for mean-field neural networks using a bounded activation function (Nitanda et al., 2022; Chizat, 2022; Chen et al., 2022; Suzuki et al., 2023b) and standard loss functions such as logistic loss and squared loss. The following is the main theorem that bounds $\frac{1}{N}\mathcal{L}^{(N)}(\mu_*^{(N)}) - \mathcal{L}(\mu_*)$. The proof is deferred to Section 4 and Appendix A.1.

**Theorem 1.** *Under Assumptions 1 and 3, it follows that*

$$\frac{\lambda}{N}\mathrm{KL}(\mu_*^{(N)}\|\mu_*^{\otimes N}) \le \frac{1}{N}\mathcal{L}^{(N)}(\mu_*^{(N)}) - \mathcal{L}(\mu_*) \le \frac{LR^2}{2N}. \tag{15}$$

A significant difference from the previous results (11), (13) with $k \to \infty$, and Kook et al. (2024) is that our bound is free from the LSI-constant. Therefore, the approximation error uniformly decreases as $N \to \infty$ at the same rate regardless of the value of LSI-constant as well as $\lambda$.

As discussed in Section 4 later, the differences between $\frac{1}{N}\mathcal{L}^{(N)}(\mu_*^{(N)})$ and $\mathcal{L}(\mu_*)$ is due to non-linearity of the loss $\ell$. In fact, since $L = 0$ for a linear loss function $\ell$, it follows that $\frac{1}{N}\mathcal{L}^{(N)}(\mu_*^{(N)}) = \mathcal{L}(\mu_*)$.

## 3.2 Application: mean-field Langevin dynamics in the finite-particle setting

As an application of Theorem 1, we present the convergence analysis of the mean-field Langevin dynamics (MFLD) in the finite-particle settings (8) and (12), sampling guarantee for the mean-field stationary distribution $\mu_* \in \mathcal{P}_2(\mathbb{R}^d)$, and uniform-in-time Wasserstein propagation of chaos.

### 3.2.1 Convergence of the mean-field Langevin dynamics

Our convergence theory assumes the logarithmic Sobolev inequality (LSI) on $\mu_*^{(N)}$.

**Assumption 4.** *There exists a constant $\bar{\alpha} > 0$ such that $\mu_*^{(N)}$ satisfies log-Sobolev inequality with constant $\bar{\alpha}$, that is, for any smooth function $g : \mathbb{R}^{dN} \to \mathbb{R}$, it follows that*

$$\mathbb{E}_{\mu_*^{(N)}}[g^2 \log g^2] - \mathbb{E}_{\mu_*^{(N)}}[g^2] \log \mathbb{E}_{\mu_*^{(N)}}[g^2] \leq \frac{2}{\bar{\alpha}} \mathbb{E}_{\mu_*^{(N)}}[\|\nabla g\|_2^2].$$

By setting $g = \sqrt{\frac{\mathrm{d}\mu^{(N)}}{\mathrm{d}\mu_*^{(N)}}}$, Assumption 4 leads to $\mathrm{KL}(\mu^{(N)} \| \mu_*^{(N)}) \leq \frac{1}{2\bar{\alpha}} \mathbb{E}_{\mu^{(N)}}\left[\left\|\nabla \log \frac{\mathrm{d}\mu^{(N)}}{\mathrm{d}\mu_*^{(N)}}\right\|_2^2\right]$.
For instance, using Holley and Stroock argument (Holley and Stroock, 1987) under the boundedness assumption $|F_0(\mathbf{x})| \leq B \ (\forall \mathbf{x} \in \mathbb{R}^{dN})$, we can verify LSI on $\mu_*^{(N)}$ with a constant $\bar{\alpha}$ that satisfies: $\bar{\alpha} \geq \frac{2\lambda'}{\lambda} \exp\left(-\frac{4NB}{\lambda}\right)$. For the detail, see Appendix B.

The following theorem demonstrates the convergence rates of $\mathcal{L}^{(N)}(\mu^{(N)})$ with the finite-particle MFLD in the continuous- and discrete-time settings. The first assertion is a direct consequence of Theorem 1 and the standard argument based on LSI for continuous-time Langevin dynamics. Whereas for the second assertion, we employ the one-step interpolation argument (Vempala and Wibisono, 2019) with some refinement to avoid the dependence on the dimensionality $dN$ where the dynamics (12) performs. The proof is given in Appendix A.2. We denote $\mu_t^{(N)} = \mathrm{Law}(\mathbf{X}_t)$ and $\mu_k^{(N)} = \mathrm{Law}(\mathbf{X}_k)$ for continuous- and discrete-time dynamics (8) and (12), respectively.

**Theorem 2.** *Suppose Assumptions 1, 3, and 4 hold. Then, it follows that*

*1. MFLD (8) in finite-particle and continuous-time setting satisfies*

$$\frac{1}{N}\mathcal{L}^{(N)}(\mu_t^{(N)}) - \mathcal{L}(\mu_*) \leq \frac{LR^2}{2N} + \exp(-2\bar{\alpha}\lambda t)\left(\frac{1}{N}\mathcal{L}^{(N)}(\mu_0^{(N)}) - \frac{1}{N}\mathcal{L}^{(N)}(\mu_*^{(N)})\right),$$

*2. MFLD (12) with $\eta\lambda' < 1/2$ in finite-particle and discrete-time setting satisfies*

$$\frac{1}{N}\mathcal{L}^{(N)}(\mu_k^{(N)}) - \mathcal{L}(\mu_*) \leq \frac{LR^2}{2N} + \frac{\delta_\eta^{(N)}}{2\bar{\alpha}\lambda} + \exp(-\bar{\alpha}\lambda\eta k)\left(\frac{1}{N}\mathcal{L}^{(N)}(\mu_0^{(N)}) - \frac{1}{N}\mathcal{L}^{(N)}(\mu_*^{(N)})\right),$$

*where $\delta_\eta^{(N)} = 16\eta(M_2^2 + \lambda'^2)(\eta M_1^2 + \lambda d) + 64\eta^2\lambda'^2(M_2^2 + \lambda'^2)\left(\frac{\mathbb{E}[\|\mathbf{X}_0\|_2^2]}{N} + \frac{1}{\lambda'}\left(\frac{M_1^2}{4\lambda'} + \lambda d\right)\right)$.*

The term of $\frac{LR^2}{2N}$ is the particle approximation error inherited from Theorem 1. Again our result shows the LSI-constant independence particle approximation error for MFLD unlike existing results (Chen et al., 2022; Suzuki et al., 2023b) where their error bounds $O(\frac{\lambda}{\alpha N})$ scale inversely with LSI-constant $\alpha$ as seen in (11) and (13). Hence, the required number of particles to achieve $\epsilon$-accurate optimization: $\frac{1}{N}\mathcal{L}^{(N)}(\mu^{(N)}) - \mathcal{L}(\mu_*) \leq \epsilon$ suggested by our result and Chen et al. (2022); Suzuki et al. (2023b) are $N = O(\frac{1}{\epsilon})$ and $N = O(\frac{\lambda}{\alpha\epsilon})$, respectively. Whereas the iterations complexity of MFLD (12) is $O(\frac{1}{\bar{\alpha}^2\lambda\epsilon}\log\frac{1}{\epsilon})$ which is same as that in Suzuki et al. (2023b) up to a difference in LSI constants $\alpha$ or $\bar{\alpha}$.

### 3.2.2 Sampling guarantee for $\mu_*$

After running the finite-particle MFLD with a sufficient number of particles for a long time, each particle is expected to be distributed approximately according to $\mu_*$. In Corollary 1, we justify this sampling procedure for $\mu_*$ as an application of Theorem 2. We set $\Delta_0^{(N)} = \frac{1}{N}\mathcal{L}^{(N)}(\mu_0^{(N)}) - \frac{1}{N}\mathcal{L}^{(N)}(\mu_*^{(N)})$ and write the marginal distribution of $\mu_t^{(N)}/\mu_k^{(N)}$ on the first particle $x^1$ as $\mu_{t,1}^{(N)}/\mu_{k,1}^{(N)}$.

**Corollary 1.** *Under the same conditions as in Theorem 2, we run MFLDs (8) and (12) with i.i.d. initial particles $\mathbf{X} = (X_0^1, \ldots, X_0^N)$. Then, it follows that*

1. *MFLD (8) in finite-particle and continuous-time setting satisfies*

$$\lambda \mathrm{KL}(\mu_{t,1}^{(N)}\|\mu_*) \le \mathcal{L}(\mu_{t,1}^{(N)}) - \mathcal{L}(\mu_*) \le \frac{LR^2}{2N} + \exp(-2\bar{\alpha}\lambda t)\Delta_0^{(N)},$$

2. *MFLD (12) with $\eta\lambda' < 1/2$ in finite-particle and discrete-time setting satisfies*

$$\lambda \mathrm{KL}(\mu_{k,1}^{(N)}\|\mu_*) \le \mathcal{L}(\mu_{k,1}^{(N)}) - \mathcal{L}(\mu_*) \le \frac{LR^2}{2N} + \frac{\delta_\eta^{(N)}}{2\bar{\alpha}\lambda} + \exp(-\bar{\alpha}\lambda\eta k)\Delta_0^{(N)}.$$

*Proof.* For any distribution $\mu^{(N)} \in \mathcal{P}_2(\mathbb{R}^{dN})$ whose marginal $\mu_i^{(N)}$ on $i$-th coordinate $x^i$ $(i \in \{1,\ldots,N\})$ are identical to each other, it follows that by the convexity of the objective function and the entropy sandwich (Nitanda et al., 2022; Chizat, 2022): $\lambda\mathrm{KL}(\mu\|\mu_*) \le \mathcal{L}(\mu) - \mathcal{L}(\mu_*)$ $(\forall\mu \in \mathcal{P}_2(\mathbb{R}^d))$,

$$\lambda\mathrm{KL}(\mu_1^{(N)}\|\mu_*) \le \mathcal{L}(\mu_1^{(N)}) - \mathcal{L}(\mu_*) \le \frac{1}{N}\mathcal{L}^{(N)}(\mu^{(N)}) - \mathcal{L}(\mu_*). \tag{16}$$

Because of i.i.d. initialization, the distributions of $\mu_t^{(N)}/\mu_k^{(N)}$ satisfies this property. That is, (16) with $\mu^{(N)} = \mu_t^{(N)}/\mu_k^{(N)}$ holds. Hence, Theorem 2 concludes the proof. □

Corollary 1 shows the convergence of the objective $\mathcal{L}(\cdot)$ and KL-divergence $\mathrm{KL}(\cdot\|\mu_*)$ which attain the minimum value at $\mu = \mu_*$. For instance, we can deduce that the particle and iteration complexities to obtain $\sqrt{\mathrm{KL}(\mu_{k,1}^{(N)}\|\mu_*)} < \epsilon$ by MFLD (12) are $O(\frac{1}{\lambda\epsilon^2})$ and $O(\frac{1}{\bar{\alpha}^2\lambda^2\epsilon^2}\log\frac{1}{\epsilon})$, respectively, whereas Kook et al. (2024) proved the following particle and iteration complexities: $O(\frac{1}{\alpha\lambda\epsilon^2})$ and $O(\frac{1}{\alpha^2\lambda^2\epsilon^2})$.

### 3.2.3 Uniform-in-time Wasserstein propagation of chaos

As another application of Theorem 2, we prove the uniform-in-time Wasserstein propagation of chaos for MFLDs (8) and (12), saying that the Wasserstein distance between finite-particle system and its mean-field limit shrinks uniformly in time as $N \to \infty$. For the mean-field limit of (8) in the continuous-time setting, we refer to (7). For the discrete-time setting (12), we define its mean-field limit as follows; let $\mu_k = \mathrm{Law}(X_k)$ be the distribution of the infinite-particle MFLD defined by

$$X_{k+1} = X_k - \eta\nabla\frac{\delta F(\mu_k)}{\delta\mu}(X_k) + \sqrt{2\lambda\eta}\xi_k, \tag{17}$$

where $\xi_k \sim \mathcal{N}(0, I_d)$. Now, the uniform-in-time Wasserstein propagation of chaos for MFLDs is given below. We set $\Delta_0^{(N)} = \frac{1}{N}\mathcal{L}^{(N)}(\mu_0^{(N)}) - \frac{1}{N}\mathcal{L}^{(N)}(\mu_*^{(N)})$ and $\Delta_0 = \mathcal{L}(\mu_0) - \mathcal{L}(\mu_*)$.

**Corollary 2.** *Suppose Assumptions 1, 2, 3, and 4 hold. Then, it follows that*

1. *discrepancy between continuous-time MFLDs (7) and (8) is uniformly bounded in time as follows:*

$$\frac{1}{N}W_2^2(\mu_t^{(N)}, \mu_t^{\otimes N}) \le \frac{4}{\alpha\lambda}\left(\frac{LR^2}{2N} + \exp(-2\bar{\alpha}\lambda t)\Delta_0^{(N)} + \exp(-2\alpha\lambda t)\Delta_0\right).$$

2. *discrepancy between discrete-time MFLDs (17) and (12) is uniformly bounded in time as follows:*

$$\frac{1}{N}W_2^2(\mu_k^{(N)}, \mu_k^{\otimes N}) \le \frac{4}{\alpha\lambda}\left(\frac{LR^2}{2N} + \frac{\delta_\eta^{(N)}}{2\bar{\alpha}\lambda} + \frac{\delta_\eta}{2\alpha\lambda} + \exp(-\bar{\alpha}\lambda\eta k)\Delta_0^{(N)} + \exp(-2\alpha\lambda\eta k)\Delta_0\right),$$

*where $\delta_\eta = 8\eta(M_2^2 + \lambda'^2)(2\eta M_1^2 + 2\lambda d) + 32\eta^2\lambda'^2(M_2^2 + \lambda'^2)\left(\mathbb{E}\left[\|X_0\|_2^2\right] + \frac{1}{\lambda'}\left(\frac{M_1^2}{4\lambda'} + \lambda d\right)\right)$.*

*Proof.* We only prove the first assertion because the second can be proven similarly. We apply the triangle inequality to $W_2$-distance as follows:

$$W_2^2(\mu_t^{(N)}, \mu_t^{\otimes N}) \leq 2\left(W_2^2(\mu_t^{(N)}, \mu_*^{\otimes N}) + W_2^2(\mu_*^{\otimes N}, \mu_t^{\otimes N})\right)$$

Note that LSI with the same constant is preserved under tensorization: $\mu_* \to \mu_*^{\otimes N}$. Then, by Taragland's inequality (Otto and Villani, 2000), Proposition 1 with $\mu = \mu_*$, and the entropy sandwich (Nitanda et al., 2022; Chizat, 2022): $\lambda\mathrm{KL}(\mu_t\|\mu_*) \leq \mathcal{L}(\mu_t) - \mathcal{L}(\mu_*)$, we get

$$\frac{\alpha}{2}W_2^2(\mu_t^{(N)}, \mu_*^{\otimes N}) \leq \mathrm{KL}(\mu_t^{(N)}\|\mu_*^{\otimes N}) \leq \frac{1}{\lambda}(\mathcal{L}^{(N)}(\mu_t^{(N)}) - N\mathcal{L}(\mu_*)),$$

$$\frac{\alpha}{2}W_2^2(\mu_*^{\otimes N}, \mu_t^{\otimes N}) = \frac{\alpha}{2}NW_2^2(\mu_*, \mu_t) \leq N\mathrm{KL}(\mu_t\|\mu_*) \leq \frac{N}{\lambda}(\mathcal{L}(\mu_t) - \mathcal{L}(\mu_*)).$$

Applying the convergence rates of finite- and infinite-particle MFLDs (Theorem 2 and Nitanda et al. (2022)), we conclude the proof. For completeness, we include the auxiliary results used in the proof in Appendix B. $\square$

Corollary 2 uniformly controls the gap between $N$-particle system $\mu_t^{(N)}/\mu_k^{(N)}$ and its mean-field limit $\mu_t^{\otimes N}/\mu_k^{\otimes N}$. Again this result shows an improved particle approximation error $O(\frac{1}{\alpha\lambda N})$ over $O(\frac{1}{\alpha^2 N})$ (Chen et al., 2022; Suzuki et al., 2023b). Additionally, the propagation of chaos result in terms of TV-norm can be proven by using Pinsker's inequality instead of Talagrand's inequality in the proof. For the continuous-time MFLDs, we get

$$\frac{1}{N}\mathrm{TV}^2(\mu_t^{(N)}, \mu_t^{\otimes N}) \leq \frac{1}{\lambda}\left(\frac{LR^2}{2N} + \exp(-2\bar{\alpha}\lambda t)\Delta_0^{(N)} + \exp(-2\alpha\lambda t)\Delta_0\right),$$

and TV-norm counterpart for the discrete-time can be derived similary.

## 4 Proof outline and key results

In this section, we provide the proof sketch of Theorem 1. Our analysis carefully treats the non-linearity of $F_0$ because the particle approximation errors, the gap between $\mathcal{L}/\mu_*$ and $\mathcal{L}^{(N)}/\mu_*^{(N)}$, come from this non-linearity. In fact if $F_0$ is a linear functional: $F_0(\mu) = \mathbb{E}_\mu[f]$ ($\exists f : \mathbb{R}^d \to \mathbb{R}$), then $\mathcal{L}^{(N)}(\mu^{(N)}) = \sum_{i=1}^N \mathbb{E}_{X^i \sim \mu_i^{(N)}}[f(X^i) + \lambda'\|X^i\|_2^2] + \lambda\mathrm{Ent}(\mu^{(N)}) \geq \sum_{i=1}^N \mathcal{L}(\mu_i^{(N)})$, where $\mu_i^{(N)}$ are marginal distributions on $X^i$. This results in $\mu_*^{(N)} = \mu_*^{\otimes N}$ and $\mathcal{L}^{(N)}(\mu_*^{(N)}) = N\mathcal{L}(\mu_*)$, and thus there is no approximation error by using finite-particles as also deduced from Theorem 1 with $L = 0$. Therefore, we should take into account the non-linearity of $F_0$ to tightly evaluate the gap between $\mathcal{L}$ and $\mathcal{L}^{(N)}$.

To do so, we define Bregman divergence based on $F$ on $\mathcal{P}_2(\mathbb{R}^d)$ as follows; for any $\mu, \mu' \in \mathcal{P}_2(\mathbb{R}^d)$,

$$B_F(\mu, \mu') = F(\mu) - F(\mu') - \left\langle \frac{\delta F(\mu')}{\delta\mu}, \mu - \mu' \right\rangle. \tag{18}$$

$B_F$ measures the discrepancy between $\mu$ and $\mu'$ in light of the strength of the convexity. If $F$ is linear with respect to the distribution, $B_F = 0$ clearly holds. By the convexity $F$, we see $B_F(\mu, \mu') \geq 0$. Moreover, we see the following relationship between $\mu_*^{(N)}$ and $\hat{\mu}$ for any $\mu \in \mathcal{P}_2(\mathbb{R}^d)$:

$$\frac{\mathrm{d}\mu_*^{(N)}}{\mathrm{d}\mathbf{x}}(\mathbf{x}) \propto \exp\left(-\frac{N}{\lambda}F(\mathbf{x})\right)$$

$$= \exp\left(-\frac{N}{\lambda}\left(F(\mu) + \left\langle\frac{\delta F(\mu)}{\delta\mu}, \mu_\mathbf{x} - \mu\right\rangle + B_F(\mu_\mathbf{x}, \mu)\right)\right)$$

$$\propto \exp\left(-\frac{N}{\lambda}B_F(\mu_\mathbf{x}, \mu)\right)\prod_{i=1}^N \exp\left(-\frac{1}{\lambda}\frac{\delta F(\mu)}{\delta\mu}(x^i)\right)$$

$$\propto \exp\left(-\frac{N}{\lambda}B_F(\mu_\mathbf{x}, \mu)\right)\frac{\mathrm{d}\hat{\mu}^{\otimes N}}{\mathrm{d}\mathbf{x}}(\mathbf{x}). \tag{19}$$

The proximal Gibbs distribution $\hat{\mu}$ has been introduced in Nitanda et al. (2022) as a proxy for the solution $\mu_*$ and it coincides with $\mu_*$ when $F$ is a linear functional. The above equation (19) naturally reflects this property since it bridges the gap between $\mu_*^{(N)}$ and $\hat{\mu}^{\otimes N}$ using $B_F$ and leads to $\mu_*^{(N)} = \hat{\mu}^{\otimes N}$ for the linear functional $F$.

Next, we provide key propositions whose proofs can be found in Appendix A. The following proposition expresses objective gaps $\mathcal{L}^{(N)}(\mu^{(N)}) - N\mathcal{L}(\mu)$ and $\mathcal{L}^{(N)}(\mu^{(N)}) - \mathcal{L}^{(N)}(\hat{\mu}^{\otimes N})$ using only divergences.

**Proposition 1.** *For $\mu \in \mathcal{P}_2(\mathbb{R}^d)$ and $\mu^{(N)} \in \mathcal{P}_2(\mathbb{R}^{dN})$, we have*

$$\mathcal{L}^{(N)}(\mu^{(N)}) - N\mathcal{L}(\mu) = N\mathbb{E}_{\mathbf{X} \sim \mu^{(N)}}\left[B_F(\mu_{\mathbf{X}}, \mu)\right] + \lambda\mathrm{KL}(\mu^{(N)}\|\hat{\mu}^{\otimes N}) - \lambda N\mathrm{KL}(\mu\|\hat{\mu}), \quad (20)$$

$$\mathcal{L}^{(N)}(\mu^{(N)}) - \mathcal{L}^{(N)}(\hat{\mu}^{\otimes N}) = N\int B_F(\mu_{\mathbf{X}}, \mu)(\mu^{(N)} - \hat{\mu}^{\otimes N})(\mathrm{d}\mathbf{x}) + \lambda\mathrm{KL}(\mu^{(N)}\|\hat{\mu}^{\otimes N}). \quad (21)$$

The following proposition shows that the KL-divergence between $\mu_*^{(N)}$ and $\hat{\mu}^{\otimes N}$ can be upper-bounded by the Bregman divergence $B_F$.

**Proposition 2.** *For any $\mu \in \mathcal{P}_2(\mathbb{R}^d)$, we have*

$$\mathrm{KL}(\mu_*^{(N)}\|\hat{\mu}^{\otimes N}) \leq \frac{N}{\lambda}\int B_F(\mu_{\mathbf{X}}, \mu)(\hat{\mu}^{\otimes N} - \mu_*^{(N)})(\mathrm{d}\mathbf{x}). \quad (22)$$

By applying Eq. (22) to Eq. (20) with $\mu = \mu_*$ and $\mu^{(N)} = \mu_*^{(N)}$, we obtain an important inequality: $\mathcal{L}^{(N)}(\mu_*^{(N)}) - N\mathcal{L}(\mu_*) \leq N\mathbb{E}_{\mathbf{X} \sim \mu_*^{\otimes N}}\left[B_F(\mu_{\mathbf{X}}, \mu_*)\right]$. Here, we give a finer result below.

**Theorem 3.** *For the minimizes $\mu_*$ of $\mathcal{L}$ and $\mu_*^{(N)}$ of $\mathcal{L}^{(N)}$, it follows that*

$$\lambda\mathrm{KL}(\mu_*^{(N)}\|\mu_*^{\otimes N}) \leq \mathcal{L}^{(N)}(\mu_*^{(N)}) - N\mathcal{L}(\mu_*)$$
$$\leq \mathcal{L}^{(N)}(\mu_*^{\otimes N}) - N\mathcal{L}(\mu_*) = N\mathbb{E}_{\mathbf{X} \sim \mu_*^{\otimes N}}\left[B_F(\mu_{\mathbf{X}}, \mu_*)\right].$$

*Proof.* Proposition 1 with $\mu = \mu_*$ and $\mu^{(N)} = \mu_*^{(N)}$ lead to the following equalities:

$$\mathcal{L}^{(N)}(\mu_*^{(N)}) - N\mathcal{L}(\mu_*) = N\mathbb{E}_{\mathbf{X} \sim \mu_*^{(N)}}\left[B_F(\mu_{\mathbf{X}}, \mu_*)\right] + \lambda\mathrm{KL}(\mu_*^{(N)}\|\mu_*^{\otimes N}), \quad (23)$$

$$\mathcal{L}^{(N)}(\mu_*^{(N)}) - \mathcal{L}^{(N)}(\mu_*^{\otimes N}) = N\int B_F(\mu_{\mathbf{X}}, \mu_*)(\mu_*^{(N)} - \mu_*^{\otimes N})(\mathrm{d}\mathbf{x}) + \lambda\mathrm{KL}(\mu_*^{(N)}\|\mu_*^{\otimes N}). \quad (24)$$

The first inequality of the theorem is a direct consequence of Eq. (23) since $B_F \geq 0$. The second inequality results from $\mathcal{L}^{(N)}(\mu_*^{(N)}) \leq \mathcal{L}^{(N)}(\mu_*^{\otimes N})$. The last equality is obtained by subtracting Eq. (24) from Eq. (23). $\qquad\square$

Intuitively, $\mathbb{E}_{\mathbf{X} \sim \mu_*^{\otimes N}}\left[B_F(\mu_{\mathbf{X}}, \mu_*)\right]$ is small because the empirical distribution $\mu_{\mathbf{X}}$ ($\mathbf{X} \sim \mu_*^{\otimes N}$) converges to $\mu_*$ by law of large numbers. Indeed, more simply, we can relate this term to the variance of an $N$-particle mean-field model $h_{\mu_{\mathbf{X}}}(z)$ ($\mathbf{X} \sim \mu_*^{\otimes N}$), yielding a bound: $\mathbb{E}_{\mathbf{X} \sim \mu_*^{\otimes N}}\left[B_F(\mu_{\mathbf{X}}, \mu_*)\right] \leq \frac{LR^2}{2N}$ (see the proof of Theorem 1 in Appendix A). Then, we arrive at Theorem 1.

## Conclusion and Discussion

We provided an improved particle approximation error over Chen et al. (2022); Suzuki et al. (2023b) by alleviating the dependence on LSI constants in their bounds. Specifically, we established an LSI-constant-free particle approximation error concerning the objective gap. Additionally, we demonstrated improved convergence of MFLD, sampling guarantee for the mean-field stationary distribution $\mu_*$, and uniform-in-time Wasserstein propagation of chaos in terms of particle complexity.

A limitation of our result is that the iteration complexity still depends exponentially on the LSI constant. This hinders achieving polynomial complexity for MFLD. However, considering the difficulty of general non-convex optimization problems, this dependency may be unavoidable. Improving the iteration complexity for more specific problem settings is an important direction for future research.

## Acknowledgment

This research is supported by National Research Foundation, Singapore and Infocomm Media Development Authority under its Trust Tech Funding Initiative, the Centre for Frontier Artificial Intelligence Research, Institute of High Performance Computing, A⋆STAR, and the College of Computing and Data Science at Nanyang Technological University. Any opinions, findings, conclusions, or recommendations expressed in this material are those of the author and do not reflect the views of National Research Foundation, Singapore, and Infocomm Media Development Authority.

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

# Appendix

## A  Omitted Proofs

### A.1  Finite-particle approximation error

In this section, we prove the LSI-constant-free particle approximation error (Theorem 1). First we give proofs of Propositions 1 and 2.

*Proof of Proposition 1.* First, we prove Eq. (20) as follows:

$$
\mathcal{L}^{(N)}(\mu^{(N)}) - N\mathcal{L}(\mu)
$$
$$
= N\mathbb{E}_{\mathbf{X}\sim\mu^{(N)}}[F(\mu_{\mathbf{X}}) - F(\mu)] + \lambda(\mathrm{Ent}(\mu^{(N)}) - N\mathrm{Ent}(\mu))
$$
$$
= N\mathbb{E}_{\mathbf{X}\sim\mu^{(N)}}\left[B_F(\mu_{\mathbf{X}}, \mu) + \left\langle \frac{\delta F}{\delta\mu}(\mu), \mu_{\mathbf{X}} - \mu \right\rangle\right] + \lambda(\mathrm{Ent}(\mu^{(N)}) - N\mathrm{Ent}(\mu))
$$
$$
= N\mathbb{E}_{\mathbf{X}\sim\mu^{(N)}}\left[B_F(\mu_{\mathbf{X}}, \mu) - \lambda\left\langle \log\frac{\mathrm{d}\hat{\mu}}{\mathrm{d}x}, \mu_{\mathbf{X}} - \mu \right\rangle\right] + \lambda(\mathrm{Ent}(\mu^{(N)}) - N\mathrm{Ent}(\mu))
$$
$$
= N\mathbb{E}_{\mathbf{X}\sim\mu^{(N)}}\left[B_F(\mu_{\mathbf{X}}, \mu) - \lambda\left\langle \log\frac{\mathrm{d}\hat{\mu}}{\mathrm{d}x}, \mu_{\mathbf{X}} \right\rangle\right] + \lambda\mathrm{Ent}(\mu^{(N)}) - \lambda N\mathrm{KL}(\mu\|\hat{\mu})
$$
$$
= N\mathbb{E}_{\mathbf{X}\sim\mu^{(N)}}[B_F(\mu_{\mathbf{X}}, \mu)] - \lambda\mathbb{E}_{\mathbf{X}\sim\mu^{(N)}}\left[\sum_{i=1}^{N}\log\frac{\mathrm{d}\hat{\mu}}{\mathrm{d}x}(X^i)\right] + \lambda\mathrm{Ent}(\mu^{(N)}) - \lambda N\mathrm{KL}(\mu\|\hat{\mu})
$$
$$
= N\mathbb{E}_{\mathbf{X}\sim\mu^{(N)}}[B_F(\mu_{\mathbf{X}}, \mu)] + \lambda\mathrm{KL}(\mu^{(N)}\|\hat{\mu}^{\otimes N}) - \lambda N\mathrm{KL}(\mu\|\hat{\mu}).
$$

Next, we prove Eq. (21) as follows:

$$
\mathcal{L}^{(N)}(\mu^{(N)}) - \mathcal{L}^{(N)}(\hat{\mu}^{\otimes N})
$$
$$
= N\int F(\mathbf{x})(\mu^{(N)} - \hat{\mu}^{\otimes N})(\mathrm{d}\mathbf{x}) + \lambda(\mathrm{Ent}(\mu^{(N)}) - \mathrm{Ent}(\hat{\mu}^{\otimes N}))
$$
$$
= N\int F(\mathbf{x})(\mu^{(N)} - \hat{\mu}^{\otimes N})(\mathrm{d}\mathbf{x}) + \lambda\mathrm{KL}(\mu^{(N)}\|\hat{\mu}^{\otimes N}) + \lambda\int \log\frac{\mathrm{d}\hat{\mu}^{\otimes N}}{\mathrm{d}\mathbf{x}}(\mathbf{x})(\mu^{(N)} - \hat{\mu}^{\otimes N})(\mathrm{d}\mathbf{x})
$$
$$
= N\int F(\mathbf{x})(\mu^{(N)} - \hat{\mu}^{\otimes N})(\mathrm{d}\mathbf{x}) + \lambda\mathrm{KL}(\mu^{(N)}\|\hat{\mu}^{\otimes N}) - \int\sum_{i=1}^{N}\frac{\delta F}{\delta\mu}(\mu)(x^i)(\mu^{(N)} - \hat{\mu}^{\otimes N})(\mathrm{d}\mathbf{x})
$$
$$
= N\int \left(F(\mathbf{x}) - \left\langle \frac{\delta F}{\delta\mu}(\mu), \mu_{\mathbf{x}} \right\rangle\right)(\mu^{(N)} - \hat{\mu}^{\otimes N})(\mathrm{d}\mathbf{x}) + \lambda\mathrm{KL}(\mu^{(N)}\|\hat{\mu}^{\otimes N})
$$
$$
= N\int \left(F(\mathbf{x}) - F(\mu) - \left\langle \frac{\delta F}{\delta\mu}(\mu), \mu_{\mathbf{x}} - \mu \right\rangle\right)(\mu^{(N)} - \hat{\mu}^{\otimes N})(\mathrm{d}\mathbf{x}) + \lambda\mathrm{KL}(\mu^{(N)}\|\hat{\mu}^{\otimes N})
$$
$$
= N\int B_F(\mu_{\mathbf{x}}, \mu)(\mu^{(N)} - \hat{\mu}^{\otimes N})(\mathrm{d}\mathbf{x}) + \lambda\mathrm{KL}(\mu^{(N)}\|\hat{\mu}^{\otimes N}).
$$

$\square$

*Proof of Proposition 2.* Let $Z_F(\mu)$ be a normalizing factor in RHS of (19), that is,

$$
Z_F(\mu) = \int \exp\left(-\frac{N}{\lambda}B_F(\mu_{\mathbf{x}}, \mu)\right)\hat{\mu}^{\otimes N}(\mathrm{d}\mathbf{x}).
$$

By the Jensen's inequality, we have

$$
\log Z_F(\mu) \geq -\int \frac{N}{\lambda}B_F(\mu_{\mathbf{x}}, \mu)\hat{\mu}^{\otimes N}(\mathrm{d}\mathbf{x}).
$$

Therefore, we get

$$
\mathrm{KL}(\mu_*^{(N)}\|\hat{\mu}^{\otimes N}) = \int \mu_*^{(N)}(\mathrm{d}\mathbf{x})\log\frac{\mathrm{d}\mu_*^{(N)}}{\mathrm{d}\hat{\mu}^{\otimes N}}(\mathbf{x})
$$

$$= \int \mu_*^{(N)}(\mathrm{d}\mathbf{x}) \log \frac{\exp\left(-\frac{N}{\lambda} B_F(\mu_\mathbf{x}, \mu)\right)}{Z_F(\mu)}$$

$$= -\int \frac{N}{\lambda} B_F(\mu_\mathbf{x}, \mu) \mu_*^{(N)}(\mathrm{d}\mathbf{x}) - \log Z_F(\mu)$$

$$\leq \frac{N}{\lambda} \int B_F(\mu_\mathbf{x}, \mu)(\hat{\mu}^{\otimes N} - \mu_*^{(N)})(\mathrm{d}\mathbf{x}).$$

$\square$

Now we are ready to prove Theorem 1.

*Proof of Theorem 1.* As discussed in Section 4, the evaluation of $\mathbb{E}_{\mathbf{X} \sim \mu_*^{\otimes N}} [B_F(\mu_\mathbf{X}, \mu_*)]$ completes the proof. For any function $G : \mathbb{R}^d \to \mathbb{R}$ so that the following integral is well defined, we have

$$\int \langle G, \mu_\mathbf{x} \rangle \, \mu_*^{\otimes N}(\mathrm{d}\mathbf{x}) = \int \frac{1}{N} \sum_{i=1}^N G(x^i) \prod_{i=1}^N \mu_*(\mathrm{d}x^i) = \int G(x) \mu_*(\mathrm{d}x) = \langle G, \mu_* \rangle.$$

Applying this equality with $G(x) = \frac{\delta F}{\delta \mu}(\mu_*)(x)$, $G(x) = \|x\|_2^2$, and $G(x) = h(x, z)$, we have

$$\int \left\langle \frac{\delta F}{\delta \mu}(\mu_*), \mu_\mathbf{x} - \mu_* \right\rangle \mu_*^{\otimes N}(\mathrm{d}\mathbf{x}) = 0,$$

$$\int \left( \mathbb{E}_{X \sim \mu_\mathbf{x}}[\|X\|_2^2] - \mathbb{E}_{X \sim \mu_*}[\|X\|_2^2] \right) \mu_*^{\otimes N}(\mathrm{d}\mathbf{x}) = 0,$$

$$\int h_{\mu_\mathbf{x}}(z) \mu_*^{\otimes N}(\mathrm{d}\mathbf{x}) = h_{\mu_*}(z).$$

Then, we can upper bound the Bregman divergence as follows.

$$N\mathbb{E}_{\mathbf{X} \sim \mu_*^{\otimes N}} [B_F(\mu_\mathbf{X}, \mu_*)]$$

$$= N\mathbb{E}_{\mathbf{X} \sim \mu_*^{\otimes N}} \left[ F(\mu_\mathbf{X}) - F(\mu_*) - \left\langle \frac{\delta F}{\delta \mu}(\mu_*), \mu_\mathbf{X} - \mu_* \right\rangle \right]$$

$$= N\mathbb{E}_{\mathbf{X} \sim \mu_*^{\otimes N}} [F_0(\mu_\mathbf{X}) - F_0(\mu_*)]$$

$$= \frac{N}{n} \sum_{j=1}^n \mathbb{E}_{\mathbf{X} \sim \mu_*^{\otimes N}} [\ell(h_{\mu_\mathbf{X}}(z_j), y_j) - \ell(h_{\mu_*}(z_j), y_j)]$$

$$\leq \frac{N}{n} \sum_{j=1}^n \mathbb{E}_{\mathbf{X} \sim \mu_*^{\otimes N}} \left[ \left. \frac{\partial \ell(a, y_j)}{\partial a} \right|_{a=h_{\mu_*}(z_j)} (h_{\mu_\mathbf{X}}(z_j) - h_{\mu_*}(z_j)) + \frac{L}{2} |h_{\mu_\mathbf{X}}(z_j) - h_{\mu_*}(z_j)|^2 \right]$$

$$= \frac{LN}{2n} \sum_{j=1}^n \mathbb{E}_{\mathbf{X} \sim \mu_*^{\otimes N}} \left[ |h_{\mu_\mathbf{X}}(z_j) - h_{\mu_*}(z_j)|^2 \right].$$

The last term is a variance of $h_{\mu_\mathbf{X}}(z)$ ($z \in \mathcal{Z}$); hence we simply evaluate it as follows.

$$\mathbb{E}_{\mathbf{X} \sim \mu_*^{\otimes N}} \left[ |h_{\mu_\mathbf{X}}(z) - h_{\mu_*}(z)|^2 \right] = \mathbb{E}_{\mathbf{X} \sim \mu_*^{\otimes N}} \left[ \left| \frac{1}{N} \sum_{i=1}^N h(X_i, z) - \mathbb{E}_{X \sim \mu_*}[h(X, z)] \right|^2 \right]$$

$$= \mathbb{E}_{\mathbf{X} \sim \mu_*^{\otimes N}} \left[ \frac{1}{N^2} \sum_{i=1}^N |h(X_i, z) - \mathbb{E}_{X \sim \mu_*}[h(X, z)]|^2 \right]$$

$$\leq \mathbb{E}_{\mathbf{X} \sim \mu_*^{\otimes N}} \left[ \frac{1}{N^2} \sum_{i=1}^N |h(X_i, z)|^2 \right]$$

$$\leq \frac{R^2}{N}.$$

Therefore, we get

$$N\mathbb{E}_{\mathbf{X}\sim\mu_*^{\otimes N}}\left[B_F(\mu_{\mathbf{X}}, \mu_*)\right] \le \frac{LR^2}{2}.$$

Combining Theorem 3, we conclude

$$\frac{1}{N}\mathcal{L}^{(N)}(\mu_*^{(N)}) - \mathcal{L}(\mu_*) \le \frac{LR^2}{2N}.$$

$\square$

## A.2 Convergence of Mean-field Langevin dynamics in the discrete-setting

In this section, we prove the convergence rate of MFLD (12). We first provide the following lemma which shows the uniform boundedness of the second moment of iterations.

**Lemma 1.** *Under Assumption 1 and $\eta\lambda' < 1/2$, we run discrete mean-field Langevin dynamics (12). Then we get*

$$\mathbb{E}[\|X_k^i\|_2^2] \le \mathbb{E}\left[\|X_0^i\|_2^2\right] + \frac{1}{\lambda'}\left(\frac{M_1^2}{4\lambda'} + \lambda d\right).$$

*Proof.* Using the inequality $(a+b)^2 \le (1+\gamma)a^2 + \left(1+\frac{1}{\gamma}\right)b^2$ with $\gamma = \frac{2\eta\lambda'}{1-2\eta\lambda'} > 0$, we have

$$\mathbb{E}\left[\|X_{k+1}^i\|_2^2\right] = \mathbb{E}\left[\left\|X_k^i - \eta\nabla\frac{\delta F(\mu_{\mathbf{X}_k})}{\delta\mu}(X_k^i) + \sqrt{2\lambda\eta}\xi_k^i\right\|_2^2\right]$$

$$= \mathbb{E}\left[\left\|(1-2\eta\lambda')X_k^i - \eta\nabla\frac{\delta F_0(\mu_{\mathbf{X}_k})}{\delta\mu}(X_k^i) + \sqrt{2\lambda\eta}\xi_k^i\right\|_2^2\right]$$

$$= \mathbb{E}\left[\left\|(1-2\eta\lambda')X_k^i - \eta\nabla\frac{\delta F_0(\mu_{\mathbf{X}_k})}{\delta\mu}(X_k^i)\right\|_2^2 + 2\lambda\eta\left\|\xi_k^i\right\|_2^2\right]$$

$$= \mathbb{E}\left[\left((1-2\eta\lambda')\left\|X_k^i\right\|_2 + \eta M_1\right)^2\right] + 2\lambda\eta d$$

$$\le (1+\gamma)(1-2\eta\lambda')^2\mathbb{E}\left[\left\|X_k^i\right\|_2^2\right] + \left(1+\frac{1}{\gamma}\right)\eta^2 M_1^2 + 2\lambda\eta d$$

$$= (1-2\eta\lambda')\mathbb{E}\left[\left\|X_k^i\right\|_2^2\right] + \eta\left(\frac{M_1^2}{2\lambda'} + 2\lambda d\right).$$

This leads to $\mathbb{E}\left[\|X_k^i\|_2^2\right] \le (1-2\eta\lambda')^k\mathbb{E}\left[\|X_0^i\|_2^2\right] + \frac{1}{\lambda'}\left(\frac{M_1^2}{4\lambda'} + \lambda d\right).$ $\square$

Now, we prove Theorem 2 that is basically an extension of one-step interpolation argument in Vempala and Wibisono (2019).

*Proof of Theorem 2.* The convergence rate in the continuous-time setting is a direct consequence of Theorem 1 and the convergence of the Langevin dynamics: $\mathcal{L}^{(N)}(\mu_t^{(N)}) \to \mathcal{L}^{(N)}(\mu_*^{(N)})$ based on LSI (Vempala and Wibisono, 2019).

Next, we prove the convergence rate in the discrete-time setting. We consider the one-step interpolation for $k$-th iteration: $X_{k+1}^i = X_k^i - \eta\nabla\frac{\delta F(\mu_{\mathbf{X}_k})}{\delta\mu}(X_k^i) + \sqrt{2\lambda\eta}\xi_k^i$, $(i \in \{1,2,\ldots,d\})$. To do so, let us consider the following stochastic differential equation: for $i \in \{1,2,\ldots,d\}$,

$$\mathrm{d}Y_t^i = -\nabla\frac{\delta F(\mu_{\mathbf{Y}_0})}{\delta\mu}(Y_0^i)\mathrm{d}t + \sqrt{2\lambda}\mathrm{d}W_t, \tag{25}$$

where $\mathbf{Y}_0 = (Y_0^1,\ldots,Y_0^d) = (X_k^1,\ldots,X_k^d)$ and $W_t$ is the standard Brownian motion in $\mathbb{R}^d$ with $W_0 = 0$. Then, the following step (26) is the solution of this equation, starting from $\mathbf{Y}_0$, at time $t$:

$$Y_t^i = Y_0^i - t\nabla\frac{\delta F(\mu_{\mathbf{Y}_0})}{\delta\mu}(Y_0^i) + \sqrt{2\lambda t}\xi^i, \quad (i \in \{1,2,\ldots,d\}), \tag{26}$$

where $\xi^i \sim \mathcal{N}(0, I_d)$ ($i \in \{1, \ldots, d\}$) are i.i.d. standard Gaussian random variables.

In this proof, we identify the probability distribution with its density function with respect to the Lebesgure measure for notational simplicity. For instance, we denote by $\mu_*^{(N)}(\mathbf{y})$ the density of $\mu_*^{(N)}$. We denote by $\nu_{0t}(\mathbf{y}_0, \mathbf{y}_t)$ the joint probability distribution of $(\mathbf{Y}_0, \mathbf{Y}_t)$ for time $t$, and by $\nu_{t|0}$, $\nu_{0|t}$ and $\nu_0$, $\nu_t$ the conditional and marginal distributions. Then, we see $\nu_0 = \mu_k^{(N)} (= \mathrm{Law}(\mathbf{X}_k))$, $\nu_\eta = \mu_{k+1}^{(N)} (= \mathrm{Law}(\mathbf{X}_{k+1}))$ (i.e., $\mathbf{Y}_\eta \overset{\mathrm{d}}{=} \mathbf{X}_{k+1}$), and

$$\nu_{0t}(\mathbf{y}_0, \mathbf{y}_t) = \nu_0(\mathbf{y}_0)\nu_{t|0}(\mathbf{y}_t|\mathbf{y}_0) = \nu_t(\mathbf{y}_t)\nu_{0|t}(\mathbf{y}_0|\mathbf{y}_t).$$

The continuity equation of $\nu_{t|0}$ conditioned on $\mathbf{y}_0$ is given as follows (Vempala and Wibisono, 2019):

$$\frac{\partial \nu_{t|0}(\mathbf{y}|\mathbf{y}_0)}{\partial t} = \nabla \cdot \left(\nu_{t|0}(\mathbf{y}|\mathbf{y}_0)N\nabla F(\mathbf{y}_0)\right) + \lambda \Delta \nu_{t|0}(\mathbf{y}|\mathbf{y}_0),$$

where we write $F(\mathbf{y}_0) = F(\mu_{\mathbf{y}_0})$ and hence $N\nabla_{\mathbf{y}^i}F(\mathbf{y}_0) = \nabla \frac{\delta F(\mu_{\mathbf{y}_0})}{\delta \mu}(y_0^i)$. Therefore, we obtain the continuity equation of $\nu_t$:

$$\begin{aligned}
\frac{\partial \nu_t(\mathbf{y})}{\partial t} &= \int \frac{\partial \nu_{t|0}(\mathbf{y}|\mathbf{y}_0)}{\partial t}\nu_0(\mathbf{y}_0)\mathrm{d}\mathbf{y}_0 \\
&= \int \left(\nabla \cdot (\nu_{0t}(\mathbf{y}_0, \mathbf{y})N\nabla F(\mathbf{y}_0)) + \lambda \Delta \nu_{0t}(\mathbf{y}_0, \mathbf{y})\right)\mathrm{d}\mathbf{y}_0 \\
&= \nabla \cdot \left(\nu_t(\mathbf{y})\int \nu_{0|t}(\mathbf{y}_0|\mathbf{y})N\nabla F(\mathbf{y}_0)\mathrm{d}\mathbf{y}_0\right) + \lambda \Delta \nu_t(\mathbf{y}) \\
&= \nabla \cdot \left(\nu_t(\mathbf{y})\left(\mathbb{E}_{\mathbf{Y}_0|\mathbf{y}}\left[N\nabla F(\mathbf{Y}_0)|\mathbf{Y}_t = \mathbf{y}\right] + \lambda \nabla \log \nu_t(\mathbf{y})\right)\right) \\
&= \lambda \nabla \cdot \left(\nu_t(\mathbf{y})\nabla \log \frac{\nu_t}{\mu_*^{(N)}}(\mathbf{y})\right) \\
&\quad + \nabla \cdot \left(\nu_t(\mathbf{y})\left(\mathbb{E}_{\mathbf{Y}_0|\mathbf{y}}\left[N\nabla F(\mathbf{Y}_0)|\mathbf{Y}_t = \mathbf{y}\right] - N\nabla F(\mathbf{y})\right)\right).
\end{aligned} \tag{27}$$

For simplicity, we write $\delta_t(\cdot) = \mathbb{E}_{\mathbf{Y}_0|}\left[N\nabla F(\mathbf{Y}_0)|\mathbf{Y}_t = \cdot\right] - N\nabla F(\cdot)$. By LSI inequality (Assumption 4) and Eq. (27), for $0 \le t \le \eta$, we have

$$\begin{aligned}
\frac{\mathrm{d}\mathcal{L}^{(N)}}{\mathrm{d}t}(\nu_t) &= \int \frac{\delta\mathcal{L}^{(N)}(\nu_t)}{\delta\mu^{(N)}}(\mathbf{y})\frac{\partial \nu_t}{\partial t}(\mathbf{y})\mathrm{d}\mathbf{y} \\
&= \lambda \int \frac{\delta\mathcal{L}^{(N)}(\nu_t)}{\delta\mu^{(N)}}(\mathbf{y})\nabla \cdot \left(\nu_t(\mathbf{y})\nabla \log \frac{\nu_t}{\mu_*^{(N)}}(\mathbf{y})\right)\mathrm{d}\mathbf{y} \\
&\quad + \int \frac{\delta\mathcal{L}^{(N)}(\nu_t)}{\delta\mu^{(N)}}(\mathbf{y})\nabla \cdot (\nu_t(\mathbf{y})\delta_t(\mathbf{y}))\,\mathrm{d}\mathbf{y} \\
&= -\lambda \int \nu_t(\mathbf{y})\nabla\frac{\delta\mathcal{L}^{(N)}(\nu_t)}{\delta\mu^{(N)}}(\mathbf{y})^\top \nabla \log \frac{\nu_t}{\mu_*^{(N)}}(\mathbf{y})\mathrm{d}\mathbf{y} \\
&\quad - \int \nu_t(\mathbf{y})\nabla\frac{\delta\mathcal{L}^{(N)}(\nu_t)}{\delta\mu^{(N)}}(\mathbf{y})^\top \delta_t(\mathbf{y})\mathrm{d}\mathbf{y} \\
&= -\lambda^2 \int \nu_t(\mathbf{y})\left\|\nabla \log \frac{\nu_t}{\mu_*^{(N)}}(\mathbf{y})\right\|_2^2 \mathrm{d}\mathbf{y} \\
&\quad - \int \nu_{0t}(\mathbf{y}_0, \mathbf{y})\lambda\nabla \log \frac{\nu_t}{\mu_*^{(N)}}(\mathbf{y})^\top (N\nabla F(\mathbf{y}_0) - N\nabla F(\mathbf{y}))\,\mathrm{d}\mathbf{y}_0\mathrm{d}\mathbf{y} \\
&\le -\lambda^2 \int \nu_t(\mathbf{y})\left\|\nabla \log \frac{\nu_t}{\mu_*^{(N)}}(\mathbf{y})\right\|_2^2 \mathrm{d}\mathbf{y} \\
&\quad + \frac{1}{2}\int \nu_{0t}(\mathbf{y}_0, \mathbf{y})\left(\lambda^2 \left\|\nabla \log \frac{\nu_t}{\mu_*^{(N)}}(\mathbf{y})\right\|_2^2 + N^2\|\nabla F(\mathbf{y}_0) - \nabla F(\mathbf{y})\|_2^2\right)\mathrm{d}\mathbf{y}_0\mathrm{d}\mathbf{y}
\end{aligned}$$

(28)

(29)

$$\leq -\frac{\lambda^2}{2}\int \nu_t(\mathbf{y})\left\|\nabla\log\frac{\nu_t}{\mu_*^{(N)}}(\mathbf{y})\right\|_2^2\,\mathrm{d}\mathbf{y} + \frac{N^2}{2}\mathbb{E}_{(\mathbf{Y}_0,\mathbf{Y})\sim\nu_{0t}}\left[\|\nabla F(\mathbf{Y}_0) - \nabla F(\mathbf{Y})\|_2^2\right]$$

$$\leq -\bar\alpha\lambda^2\mathrm{KL}(\nu_t\|\mu_*^{(N)}) + \frac{N^2}{2}\mathbb{E}_{(\mathbf{Y}_0,\mathbf{Y})\sim\nu_{0t}}\left[\|\nabla F(\mathbf{Y}_0) - \nabla F(\mathbf{Y})\|_2^2\right]$$

$$= -\bar\alpha\lambda\left(\mathcal{L}^{(N)}(\nu_t) - \mathcal{L}^{(N)}(\mu_*^{(N)})\right) + \frac{N^2}{2}\mathbb{E}_{(\mathbf{Y}_0,\mathbf{Y})\sim\nu_{0t}}\left[\|\nabla F(\mathbf{Y}_0) - \nabla F(\mathbf{Y})\|_2^2\right].$$
$$(30)$$

Next, we bound the last term as follows:

$$N^2\mathbb{E}_{(\mathbf{Y}_0,\mathbf{Y})\sim\nu_{0t}}\left[\|\nabla F(\mathbf{Y}_0) - \nabla F(\mathbf{Y})\|_2^2\right]$$

$$= \mathbb{E}_{(\mathbf{Y}_0,\mathbf{Y})\sim\nu_{0t}}\left[\sum_{i=1}^N\left\|\nabla\frac{F(\mu_{\mathbf{Y}_0})}{\delta\mu}(Y_0^i) - \nabla\frac{F(\mu_{\mathbf{Y}})}{\delta\mu}(Y^i)\right\|_2^2\right]$$

$$\leq 2\mathbb{E}_{(\mathbf{Y}_0,\mathbf{Y})\sim\nu_{0t}}\left[\sum_{i=1}^N\left\{\left\|\nabla\frac{F_0(\mu_{\mathbf{Y}_0})}{\delta\mu}(Y_0^i) - \nabla\frac{F_0(\mu_{\mathbf{Y}})}{\delta\mu}(Y^i)\right\|_2^2 + 4\lambda'^2\left\|Y_0^i - Y^i\right\|_2^2\right\}\right]$$

$$\leq 4\mathbb{E}_{(\mathbf{Y}_0,\mathbf{Y})\sim\nu_{0t}}\left[NM_2^2W_2^2(\mu_{\mathbf{Y}_0},\mu_{\mathbf{Y}}) + (M_2^2 + 2\lambda'^2)\sum_{i=1}^N\left\|Y_0^i - Y^i\right\|_2^2\right]$$

$$\leq 8(M_2^2 + \lambda'^2)\mathbb{E}_{(\mathbf{Y}_0,\mathbf{Y})\sim\nu_{0t}}\left[\sum_{i=1}^N\left\|Y_0^i - Y^i\right\|_2^2\right]$$

$$\leq 8(M_2^2 + \lambda'^2)\mathbb{E}_{\mathbf{Y}_0,(\xi^i)_{i=1}^N}\left[\sum_{i=1}^N\left\|-t\nabla\frac{\delta F(\mu_{\mathbf{Y}_0})}{\delta\mu}(Y_0^i) + \sqrt{2\lambda t}\xi^i\right\|_2^2\right]$$

$$\leq 8(M_2^2 + \lambda'^2)\mathbb{E}_{\mathbf{Y}_0,(\xi^i)_{i=1}^N}\left[t^2\sum_{i=1}^N\left\|\nabla\frac{\delta F(\mu_{\mathbf{Y}_0})}{\delta\mu}(Y_0^i)\right\|_2^2 + 2\lambda t\sum_{i=1}^N\left\|\xi^i\right\|_2^2\right]$$

$$\leq 8(M_2^2 + \lambda'^2)\mathbb{E}_{\mathbf{Y}_0}\left[t^2\sum_{i=1}^N 2\left(\left\|\nabla\frac{\delta F_0(\mu_{\mathbf{Y}_0})}{\delta\mu}(Y_0^i)\right\|^2 + 4\lambda'^2\left\|Y_0^i\right\|_2^2\right) + 2\lambda tNd\right]$$

$$\leq 16N(M_2^2 + \lambda'^2)(t^2M_1^2 + \lambda td) + 64t^2\lambda'^2(M_2^2 + \lambda'^2)\mathbb{E}_{\mathbf{Y}_0}\left[\|\mathbf{Y}_0\|_2^2\right]$$

$$\leq 16N(M_2^2 + \lambda'^2)(t^2M_1^2 + \lambda td) + 64t^2\lambda'^2(M_2^2 + \lambda'^2)\left(\mathbb{E}\left[\|\mathbf{X}_0\|_2^2\right] + \frac{N}{\lambda'}\left(\frac{M_1^2}{4\lambda'} + \lambda d\right)\right).$$

Therefore for any $t\in[0,\eta]$, we see $N^2\mathbb{E}_{(\mathbf{Y}_0,\mathbf{Y})\sim\nu_{0t}}\left[\|\nabla F(\mathbf{Y}_0) - \nabla F(\mathbf{Y})\|_2^2\right] \leq N\delta_\eta^{(N)}$, where $\delta_\eta^{(N)} = 16\eta(M_2^2 + \lambda'^2)(\eta M_1^2 + \lambda d) + 64\eta^2\lambda'^2(M_2^2 + \lambda'^2)\left(\frac{1}{N}\mathbb{E}\left[\|\mathbf{X}_0\|_2^2\right] + \frac{1}{\lambda'}\left(\frac{M_1^2}{4\lambda'} + \lambda d\right)\right)$.

Substituting this bound into Eq. (30), we get for $t\in[0,\eta]$,

$$\frac{\mathrm{d}}{\mathrm{d}t}\left(\mathcal{L}^{(N)}(\nu_t) - \mathcal{L}^{(N)}(\mu_*^{(N)}) - \frac{N\delta_\eta^{(N)}}{2\bar\alpha\lambda}\right) \leq -\bar\alpha\lambda\left(\mathcal{L}^{(N)}(\nu_t) - \mathcal{L}(\mu_*^{(N)}) - \frac{N\delta_\eta^{(N)}}{2\bar\alpha\lambda}\right).$$

Noting $\nu_\eta = \mu_{k+1}^{(N)}$ and $\nu_0 = \mu_k^{(N)}$, the Gronwall's inequality leads to

$$\mathcal{L}^{(N)}(\mu_{k+1}^{(N)}) - \mathcal{L}^{(N)}(\mu_*^{(N)}) - \frac{N\delta_\eta^{(N)}}{2\bar\alpha\lambda} \leq \exp(-\bar\alpha\lambda\eta)\left(\mathcal{L}^{(N)}(\mu_k^{(N)}) - \mathcal{L}^{(N)}(\mu_*^{(N)}) - \frac{N\delta_\eta^{(N)}}{2\bar\alpha\lambda}\right).$$

This inequality holds at every iteration of (25). Hence, we arrive at the desired result,

$$\mathcal{L}^{(N)}(\mu_k^{(N)}) - \mathcal{L}^{(N)}(\mu_*^{(N)}) \leq \frac{N\delta_\eta^{(N)}}{2\bar\alpha\lambda} + \exp(-\bar\alpha\lambda\eta k)\left(\mathcal{L}^{(N)}(\mu_0^{(N)}) - \mathcal{L}^{(N)}(\mu_*^{(N)}) - \frac{N\delta_\eta^{(N)}}{2\bar\alpha\lambda}\right).$$

$$\square$$

# B   Auxiliary results

In this section, we showcase auxiliary results used in our theory.

LSI on $\mu_*^{(N)}$ can be verified by using the following two lemmas. Lemma 2 says that the strong log-concave densities satisfy the LSI with a dimension-free constant.

**Lemma 2** (Bakry and Émery (1985)). *Let $\frac{\mathrm{d}\mu(x)}{\mathrm{d}x} \propto \exp(-f(x))$ be a probability density, where $f : \mathbb{R}^d \to \mathbb{R}$ is a smooth function. If there exists $c > 0$ such that $\nabla^2 f \succeq cI_d$, then $\mu$ satisfies log-Sobolev inequality with constant $c$.*

Additionally, the LSI is preserved under bounded perturbation as seen in Lemma 3.

**Lemma 3** (Holley and Stroock (1987)). *Let $\mu$ be a probability distribution on $\mathbb{R}^d$ satisfying the log-Sobolev inequality with a constant $\alpha$. For a bounded function $B : \mathbb{R}^d \to \mathbb{R}$, we define a probability distribution $\mu_B$ as follows:*

$$\frac{\mathrm{d}\mu_B(x)}{\mathrm{d}x} = \frac{\exp(B(x))}{\mathbb{E}_{X\sim\mu}[\exp(B(X))]} \frac{\mathrm{d}\mu(x)}{\mathrm{d}x}.$$

*Then, $\mu_B$ satisfies the log-Sobolev inequality with a constant $\alpha / \exp(4\|B\|_\infty)$.*

Theorems 4 and 5 give convergence rates of the infinite-particle MFLDs in continuous- and discrete-time settings.

**Theorem 4** (Nitanda et al. (2022)). *Under Assumptions 1 and 2, we run the infinite-particle MFLD (7) in the continuous-time setting. Then, it follows that*

$$\mathcal{L}(\mu_t) - \mathcal{L}(\mu_*) \leq \exp(-2\alpha\lambda t)\left(\mathcal{L}(\mu_0) - \mathcal{L}(\mu_*)\right).$$

For MFLD (17), we consider one-step interpolation: for $0 \leq t \leq \eta$,

$$X_{k,t} = X_k - t\nabla\frac{\delta F(\mu_k)}{\delta\mu}(X_k) + \sqrt{2\lambda t}\xi_k.$$

Set $\mu_{k,t} = \mathrm{Law}(X_{k,t})$ and $\delta_{\mu_k,t} = \mathbb{E}_{(X_k, X_{k,t})}\left[\left\|\nabla\frac{\delta F(\mu_k)}{\delta\mu}(X_k) - \nabla\frac{\delta F(\mu_{k,t})}{\delta\mu}(X_{k,t})\right\|_2^2\right]$.

**Theorem 5** (Nitanda et al. (2022)). *Under Assumptions 1 and 2, we run the infinite-particle MFLD (17) in the discrete-time setting with the step size $\eta$. Suppose there exists a constant $\delta_\eta$ such that $\delta_{\mu_k,t} \leq \delta_\eta$ for any $0 < t \leq \eta$. Then, it follows that*

$$\mathcal{L}(\mu_k) - \mathcal{L}(\mu_*) \leq \frac{\delta_\eta}{2\alpha\lambda} + \exp(-\alpha\lambda\eta k)\left(\mathcal{L}(\mu_0) - \mathcal{L}(\mu_*)\right).$$

Under Assumption 1, we can evaluate $\delta_{\mu_k,t}$ in a similar way as the proof of Theorem 2 and we obtain

$$\delta_\eta = 8\eta(M_2^2 + \lambda'^2)(2\eta M_1^2 + 2\lambda d) + 32\eta^2\lambda'^2(M_2^2 + \lambda'^2)\left(\mathbb{E}\left[\|X_0\|_2^2\right] + \frac{1}{\lambda'}\left(\frac{M_1^2}{4\lambda'} + \lambda d\right)\right).$$

The next theorem gives a relationship between LSI and Talagrand's inequalities.

**Theorem 6** (Otto and Villani (2000)). *If a probability distribution $\mu \in \mathcal{P}_2(\mathbb{R}^d)$ satisfies the log-Sobolev inequality with constant $\alpha > 0$, then $\mu$ satisfies Talagrand's inequality with the same constant: for any $\mu' \in \mathcal{P}_2(\mathbb{R}^d)$*

$$\frac{\alpha}{2}W_2^2(\mu', \mu) \leq \mathrm{KL}(\mu'\|\mu),$$

*where $W_2(\mu', \mu)$ denotes the 2-Wasserstein distance*

