# OpenReview forum: "Improved Particle Approximation Error for Mean Field Neural Networks"
_NeurIPS.cc/2024/Conference — NeurIPS 2024 poster_

### Official Review · Reviewer_4H1M · 2024-06-25

**Soundness:** 3
**Presentation:** 3
**Contribution:** 3
**Rating:** 7
**Confidence:** 4

**Summary:**

The manuscript investigates the mean-field Langevin dynamics (MFLD) and improves the particle approximation error by removing the dependency on the log-Sobolev inequality (LSI) constant. This is of relevance, as the LSI constant in general might deteriorate with the regularization coefficient, limiting the impact of prior bounds.

The Authors illustrate the applicability of their result with three examples.
1. An improved convergence of the MFLD,
2. sampling guarantees for the final stationary distribution,
3. uniform-in-time propagation of chaos w.r.t. the Wasserstein distance for the MFLD.

**Strengths:**

- Improving the estimates of the particle approximation of the MFLD is of interest given its appearance in the learning problem of mean-field neural networks, which has been an interesting research topic pursued by different research groups during the last years. The paper is of purely theoretical nature and improves upon the state-of-the-art by removing the dependence of the approximation error constant on the LSI constant. The technical tools to do so, seem novel.

- The paper is well-written and -structured with the objectives and contribution clearly stated and pursued.

- In my opinion, the content of the paper could be also of interest beyond the scope of the MFLD, as propagation of chaos results with favorable constants are of interest in a wide variety of fields. The Authors may want to consider commenting on this.

**Weaknesses:**

Apart from some minor questions addressed below, there is one point of critique that I would like to raise.
Namely, the Authors do not really motivate why one should expect that the particles approximation _does not_ depend on the LSI constant. This, in my opinion, would improve the reading experience of the paper.

**Questions:**

- line 61: How does the LSI constant $\alpha$ deteriorate with the regularization parameter $\lambda$? Could you provide a sketch to give some more information? This could be also worth to be included and referenced in the manuscript.

And some more minor comments:

- In line 28, the Authors may want to add one further line of work, namely "Mean field analysis of neural networks: A law of large numbers" and "Mean field analysis of neural networks: A central limit theorem" by J Sirignano, K Spiliopoulos, to exhaustively cover the literature.
- In regards of (2), the Authors might want to mention that $\nabla \frac{\delta \mathcal{F}}{\delta \mu}$ is also known as the Wasserstein gradient $\nabla_W \mathcal{F}$.
- lines 35, 36: Could you add references here?
- lines 99, 100: Write $\mathcal{P}(\mathbb{R}^d)$.
- line 100: Not sure what you mean with "it follwos that" here
- line 157: $N$ not $d$
- line 234: identical to
- lines 266-268: This sentence sounds a bit complicated.
- line 301: Discussion

**Limitations:**

The Authors do point out some limitations in the Conclusions, which are reasonable and in my opinion justifiable.

They could further emphasize some limitations of their result following from Assumption 3. In particular the uniform boundedness of $h$ seems to be a restriction, but since the class of covered neural networks is still fair, I would not consider it as a substantial limitation. Yet, it could be highlighted.

---

> ### Author Rebuttal · Authors · 2024-08-06
>
> Thank you for the positive feedback and useful suggestions.
>
> **The Authors do not really motivate why one should expect that the particles approximation does not depend on the LSI constant.**
>
> The particle approximation error is due to the nonlinearity of $F_0(\mu)$ with respect to $\mu$. In fact, Langevin dynamics, a special case of MFLD for the linear functional $F_0$, can be simulated with only one particle. On the other hand, LSI-constant is affected by the regularization strength $\lambda, \lambda'$ and the boundedness or Lipschitz continuity of $F_0( \mu_\mathbf{x})$ or $\delta F_0( \mu_\mathbf{x}) /\delta \mu$ w.r.t $\mathbf{x}$. For instance, in the learning mean-field neural network setting (Eq. (14)), LSI is satisfied as long as the activation $h$ is bounded. In this way, these two concepts (approximation error and LSI) come from different factors and do not seem to have a direct relationship, and hence we expect that an LSI-constant free approximation error can hold.
>
>
> **Q: line 61: How does the LSI constant deteriorate with the regularization parameter? Could you provide a sketch to give some more information? This could be also worth to be included and referenced in the manuscript.**
>
> Let us consider the Gibbs distribution proportional to $\exp( f(x) - g(x) )$, where $g$ is $c$-strongly convex and $f$ is uniformly bounded by $B$.
> Then, combining techniques of [Bakry and Émery (1985)] and [Holley and Stroock (1987)], we can prove LSI with an estimated LSI constant: $c \exp(-4B)$. Please also refer to Appendix A.2 of [Nitanda, A., Wu, D., and Suzuki, T. (2021)].
> In our case of the proximal Gibbs distribution, $f$ and $g$ correspond to $- \frac{1}{\lambda} \frac{\delta F_0(\mu)}{\delta \mu}$ and $g = \lambda' \|x\|\_2^2$, respectively. Hence we can conclude that LSI constant is $\frac{2\lambda' }{\lambda} \exp(- 4B/\lambda )$, as described on line 61 if $\|\frac{\delta F_0(\mu)}{\delta \mu}\|_\infty \leq B$. This estimation is often used in the literature. For instance, please see [Nitanda et al., 2022; Chizat, 2022; Suzuki et al. 2023b].
> Moreover, $f$ could be Lipschitz continuous function because we can reformulate "the sum of Lipschitz continuous function and strongly convex function" into "the sum of bounded function and strongly convex function" by using Miclo's trick [Bardet et al., 2018].
> We will mention these techniques in Appendix.
>
> [Bakry, D. and Émery, M. (1985)] Diffusions hypercontractives. In Seminaire de probabilités XIX 1983/84,
> pages 177–206. Springer, 1985.
>
> [Holley, R. and Stroock, D. (1987)] Logarithmic Sobolev inequalities and stochastic ising models. Journal of statistical physics, 46(5-6):1159–1194
>
> **Other suggestions**
>
> Thank you for your thorough reading. We will update the manuscript and fix typos according to your comments.

---

> > ### Comment · Reviewer_4H1M · 2024-08-08
> >
> > I thank the Authors for their reply. Since my comments were sufficiently answered, I remain with my initial positive evaluation.

---

### Official Review · Reviewer_Hk29 · 2024-07-03

**Soundness:** 4
**Presentation:** 2
**Contribution:** 2
**Rating:** 6
**Confidence:** 4

**Summary:**

The paper presents an improved finite particle approximation error bound for mean-field Langevin dynamics. The main result establishes a $O(1/N)$ gap between the original and $N$-particle objective which is independent of the LSI constant and improves upon the existing $O(\lambda/\alpha N)$ bound. The is applied to obtain improved convergence error and Wasserstein propagation of chaos of MFLD for shallow neural networks.

**Strengths:**

The independence of the particle discretization error on the LSI constant is a strong and useful result, especially in low temperature regimes where $\alpha$ can be exponentially large w.r.t. $\lambda^{-1}$. The proof technique of introducing the induced Bregman divergence to bound nonlinearity is both novel and simpler compared to previous analyses. The new error is easily incorporated into existing MFLD frameworks.

**Weaknesses:**

* Besides Theorem 1, the remaining results seem to be a straightforward application to the existing analysis of time discretization of MFLD. In particular, the $\exp(-\alpha\lambda\eta k)$ convergence rate and $\eta/\alpha\lambda$ error term of [1] remain and hence the overall rate and error bound still suffer from the LSI constant, albeit decoupled from particle error.
* The corresponding Wasserstein error bound also automatically suffers from the LSI constant due to using Talagrand's inequality. While the constant is improved from $\alpha^{-2}$ to $\alpha^{-1}$, the difference seems to be incremental as $\alpha$ is already adversarially exponentially dependent on $\lambda^{-1}$.
* The standard assumptions of MFLD apply, for example the activation must be Lipschitz smooth as well as bounded to ensure that Assumption 1 holds. These implications should be explained in the text alongside Assumption 3 as currently Assumption 1 is only given for an abstract functional and not for the specific form (14).

[1] Suzuki et al, 2023. Convergence of mean-field Langevin dynamics: Time and space discretization, stochastic gradient, and variance reduction.

**Questions:**

* Currently the dependence of the convergence rate on $\alpha$ in the MFLD framework may be unavoidable, however is there a possibility that the time discretization error or Wasserstein error can also be made independent of the LSI constant?

**Limitations:**

The authors address limitations.

---

> ### Author Rebuttal · Authors · 2024-08-06
>
> Thank you for the positive feedback and helpful comments.
>
> **Besides Theorem 1, the remaining results seem to be a straightforward application. The convergence rate $\exp(-\alpha \lambda\eta k)$ and error bound $\eta/\alpha \lambda$ still suffer from the LSI constan.**
>
> As the reviewer pointed out, the results in Section 3.2 are basically obtained by the combination and modification of existing results. However, we would like to emphasize that our main aim is to derive an improved particle approximation error, and the results in Section 3.2 serve to demonstrate the effectiveness of our result. The fact that we can improve the approximation error of existing results by combining them with our result would be a point worth noting.
>
> **The Wasserstein error bound suffers from the LSI constant. Q: however is there a possibility that the time discretization error or Wasserstein error can also be made independent of the LSI constant?**
>
> At the moment, it is nontrivial whether the time discretization error of MFLD and Wasserstein propagation chaos can be independent of the LSI constant. However, it is worth mentioning that we can derive a propagation of chaos regarding TV norm, which is free from the LSI constant. This can be proven by replacing Talagrand's inequality with Pinsker's inequality $\mathrm{TV}(\mu , \mu') \leq \sqrt{2 \mathrm{KL}(\mu \| \mu')}$. The convergence in terms of TV norm is also important as it implies the weak convergence of distributions. We will add this comment in the revision.
>
>
> **The standard assumptions of MFLD apply, for example the activation must be Lipschitz smooth as well as bounded to ensure that Assumption 1 holds. These implications should be explained in the text alongside Assumption 3.**
>
> Thank you for the suggestion. Indeed, the smoothness and boundedness of the activation function can be used for verifying Assumption 1. We will update the manuscript accordingly.

---

> > ### Comment · Reviewer_Hk29 · 2024-08-07
> >
> > Thank you for the reply. I will maintain my positive assessment of the paper.

---

### Official Review · Reviewer_FiBA · 2024-07-13

**Soundness:** 4
**Presentation:** 4
**Contribution:** 3
**Rating:** 7
**Confidence:** 3

**Summary:**

This paper provides an improved bound on the particle approximation error of MFLD under conditions of a log-Sobolev inequality (and a couple of extra regularity conditions). This bound improves on previous bounds by removing the dependence of the bound on the LSI constant. The authors then apply their bound to get improved convergence rates of MFLD in the finite-particle setting, sampling guarantees for $\mu_*$ and a uniform-in-time propagation of chaos result analagous to previous literature.

**Strengths:**

- This work addresses an interesting problem - the convergence of MFLD - and provides a clear improvement over the previous best bounds in this domain. The proof method is also novel.
- The work is extremely clearly presented, with all of the key contributions being well-motivated and explained. The relevant literature is also well summarized.
- Overall, I consider this to be a very sound piece of theoretical work.

**Weaknesses:**

I have no major concerns about this work. The bound consists only of a quantitative improvement on existing bounds, rather than a completely novel bound, and still requires the same log-Sobolev assumptions as previous works, perhaps limiting its general applicability. In addition, they also assume bounded activation functions, which is an additional restriciton. However, I consider both of these restrictions to be minor.

**Questions:**

I have no clarifications to request.

**Limitations:**

The authors do provide a (quite brief) discussion of the limitations of their work in the final section. Beyond this, I foresee no other potential negative societal impacts of their work.

---

> ### Author Rebuttal · Authors · 2024-08-06
>
> Thank you for the positive feedback and thoughtful comments.
>
> **About bounded activations**
>
> In fact, the boundedness of the activation function is commonly assumed in the literature. Hence, it is not a critical limitation. That being said, we can relax the boundedness for each factor. For instance, the boundedness of $h(\cdot,z)$ can be replaced with Lipschitz continuity for deriving the LSI constant and can be replaced with the bounded second moment of $h(X,z)~(X\sim \mu_*)$ for evaluating Bregman divergence. The most critical point to be careful about is the relaxation for Assumption 1. Indeed, Assumption 1 with an unbounded activation function imposes another limitation: the output layer must be fixed, and the derivative of the loss function $\ell$ must be bounded (e.g., logistic loss).

---

> > ### Comment · Reviewer_FiBA · 2024-08-07
> >
> > Thank you for the additional information - I think my original assessment is unchanged.

---

### Official Review · Reviewer_hVSM · 2024-07-13

**Soundness:** 2
**Presentation:** 3
**Contribution:** 3
**Rating:** 6
**Confidence:** 4

**Summary:**

This work proves a novel particle approximation error for the continuous- and discrete-time mean-field Langevin dynamics (MFLD), which removes the dependency on the log-Sobolev inequality (LSI) constant in the number of particles, leading to a potential exponential in inverse-temperature improvement in the number of particles required. This new bound is used to improve prior optimization, sampling, and uniform-in-time propagation of chaos guarantees for the MFLD.

**Strengths:**

The Bregman divergence based technique of analyzing the particle discretization error is novel and interesting, and can inspire further research in this line of work. The intuition behind the proof is also nicely explained in the main text.

**Weaknesses:**

Currently my main concern is the verification of Assumption 4 in standard settings. The authors cite Lemma 22 of Kook et al., 2024 to obtain a uniform-in-N LSI constant for ${\mu_*}^{(N)}$. However, in the new version of Kook et al., 2024, Lemma 22 only proves an LSI constant for the proximal Gibbs measure. In the new version, Corollary 25 does prove a uniform-in-N LSI constant for $\mu^{(N)}_*$, but this requires $N$ to be exponentially large in the inverse temperature $1/\lambda$, which is what this paper tried to avoid. I would be happy to increase my score if this issue is resoled.

Additionally, minor comments and questions are provided in the “Questions” section below.

**Questions:**

* What is the main challenge in handling unbounded activations? Furthermore, does the boundedness of the first variation in Assumption 1 play an important role, or can it be removed as the bound does not enter any of the estimates?
* In Line 162, the authors refer to Equation (11) as the continuous-time propagation of chaos error, while Equation (11) only considers the error at optimality, and the continuous-time propagation of chaos result is introduced in Theorem 2. This might need more clarification.
* In Assumption 3, $\ell$ is said to be Lipschitz smooth, while in fact its gradient is assumed to be Lipschitz. It might be more clear to refer to $\ell$ as simply $L$-smooth.
* The step size appearing on the RHS in Part 1 of Theorem 2 might be a typo, since it is a continuous-time statement.

**Limitations:**

The authors have adequately discussed the limitations of their work. The main limitation is the fact that the number of iterations and therefore the computational complexity still depend on the LSI constant, which make the algorithm inefficient in low noise regimes in the worst case. Another limitation is assuming bounded activations, which could potentially be relaxed with a more detailed analysis.

---

> ### Author Rebuttal · Authors · 2024-08-06
>
> Thank you for your careful reading and feedback.
>
> **Verification of Assumption 4 and Lemma 22 of [Kook et al., 2024]**
>
> Thank you for bringing up this point. Indeed, the authors of [Kook et al., 2024] updated their manuscript due to an error in Lemma 22 of the previous version. As a result, a new $N$-independent LSI constant (Corollary 25 of [Kook et al., 2024]) requires a strong assumption on $N$. Therefore, we plan to simply replace it with the LSI constant: $\alpha \geq \frac{\lambda'}{\lambda}\exp(-NB/\lambda)$, supported by Holley-Stroock argument under boundedness of $F_0$. Although this LSI constant depends on $N$, it is useful in practice because we basically want to use the minimum number of $N$ to achieve a given accuracy.
>
> In fact, an LSI constant $\frac{\lambda'}{\lambda}\exp(-NB/\lambda)$ is rather better than the original constant in between lines 204-205 of the submission when $N \leq 1/\lambda'$, which aligns with practical machine learning settings. Specifically, for a given number of examples $n$ and $\gamma>0$ (e.g., $\gamma=1/2, 1$), we usually set $\lambda' = 1/n^\gamma$, and our particle approximation bound also suggests an error of $1/n^\gamma$ when $N=n^\gamma$, which indeed satisfies $N \leq 1/\lambda'$. In other words, under this typical machine learning setting, the estimated LSI-constant $\alpha \geq \frac{\lambda'}{\lambda}\exp(-NB/\lambda)$ is better than the LSI constant (lines 204-205) based on the previous version of [Kook et al., 2024].
>
> Moreover, we confirmed that our proof technique can be used to improve their LSI result (Corollary 25 of [Kook et al., 2024]) so that it holds any $N$. Indeed, the assumption on $N$ of Corollary 25 stems from Wasserstein propagation of chaos (Theorem 26), which inherently requires $N \geq 1/\alpha$ (see also Theorem 5). On the other hand, our propagation of chaos result holds for any $N$, and hence, we can remove this condition from Theorem 26. However, our aim is not to improve the LSI but to derive an LSI-constant free particle approximation error. Therefore, the improvement of the LSI will be studied in future work with further refinement, and we will just utilize $N$-dependent LSI for our submission.
>
> **The main challenge in handling unbounded activations**
>
> The boundedness is due to several factors: Lipschitz smoothness of the first variation (Assumption 1), the LSI-constant by Holley-Strook argument, and the evaluation of Bregman divergence. That being said, we can relax the boundedness for each factor. For instance, the boundedness of $h(\cdot,z)$ can be replaced with the Lipschitz continuity for deriving the LSI-constant and with the bounded second moment of $h(X,z)~(X\sim \mu_*)$ for evaluating Bregman divergence. The most critical point to be careful about is the relaxation for Assumption 1. Indeed, Assumption 1 with an unbounded activation function introduces another limitation: the output layer must be fixed, and the derivative of the loss function $\ell$ must be bounded (e.g., logistic loss).
>
> We also would like to note that the boundedness of the first variation in Assumption 1 simply says the differentiability of $F_0$. This seems redundant, so we will omit it in the revision.
>
> **In Line 162, the authors refer to Equation (11) as the continuous-time propagation of chaos error, while Equation (11) only considers the error at optimality**
>
> Thank you for pointing it out. Indeed, Eq. (11) only considers the error at the solution. We will simply reward this sentence as follows: "The finite-particle approximation error $O(\frac{\lambda}{\alpha N})$ appearing in (11), (13) means ..."
>
> Moreover, we will update the manuscript according to your suggestion and fix the typo. Thank you for pointing them out.

---

> > ### Comment · Reviewer_hVSM · 2024-08-11
> >
> > Thank you for your detailed responses and clarifications. My concern about the LSI constant estimate after Assumption 4 is resolved, therefore I have increased my rating to 6.

---

### Decision · Program_Chairs · 2024-09-25

**Decision:**

Accept (poster)

**Comment:**

This paper presents an improved bound on the particle approximation error for both continuous- and discrete-time mean-field Langevin dynamics in the context of mean-field neural networks. The result is established under conditions considered in previous works, namely a log-Sobolev inequality (LSI) condition and additional regularity assumptions. The key improvement is that the bound in the continuous setting no longer depends on the LSI constant, in contrast to existing results. The authors then applied this new bound to improve convergence rates in the finite-particle setting, provide sampling guarantees for the stationary distribution, and establish a uniform-in-time propagation of chaos result.

All the reviewers and I agree that the paper is a valuable addition to the literature and should be accepted at Neurips.